# A microRNA negative feedback loop downregulates vesicle transport and inhibits fear memory

Rebecca S Mathew[1†], Antonis Tatarakis[1†], Andrii Rudenko[2‡], Erin M Johnson-Venkatesh[3], Yawei J Yang[4,5,6], Elisabeth A Murphy[4], Travis P Todd[7], Scott T Schepers[7], Nertila Siuti[1], Anthony J Martorell[2], William A Falls[7], Sayamwong E Hammack[7], Christopher A Walsh[4,5,8], Li-Huei Tsai[2,8], Hisashi Umemori[3], Mark E Bouton[7], Danesh Moazed[1*]

[1]Department of Cell Biology, Howard Hughes Medical Institute, Harvard Medical School, Boston, United States; [2]Department of Brain and Cognitive Sciences Massachusetts Institute of Technology, The Picower Institute for Learning and Memory, Cambridge, United States; [3]Department of Neurology, FM Kirby Neurobiology Center, Boston Children's Hospital, Harvard Medical School, Boston, United States; [4]Division of Genetics, Howard Hughes Medical Institute, Manton Center for Orphan Disease Research, Boston Children's Hospital, Boston, United States; [5]Program in Biological and Biomedical Sciences, Harvard Medical School, Boston, United States; [6]Harvard-MIT Division of Health Sciences and Technology, Harvard Medical School, Boston, United States; [7]Department of Psychology, University of Vermont, Burlington, United States; [8]Broad Institute of MIT and Harvard, Cambridge, United States

*For correspondence: danesh@hms.harvard.edu

†These authors contributed equally to this work

Present address: ‡Department of Biology, The City College of the City University of New York, New York, United States

Competing interests: The authors declare that no competing interests exist.

**Abstract** The SNARE-mediated vesicular transport pathway plays major roles in synaptic remodeling associated with formation of long-term memories, but the mechanisms that regulate this pathway during memory acquisition are not fully understood. Here we identify miRNAs that are up-regulated in the rodent hippocampus upon contextual fear-conditioning and identify the vesicular transport and synaptogenesis pathways as the major targets of the fear-induced miRNAs. We demonstrate that miR-153, a member of this group, inhibits the expression of key components of the vesicular transport machinery, and down-regulates Glutamate receptor A1 trafficking and neurotransmitter release. MiR-153 expression is specifically induced during LTP induction in hippocampal slices and its knockdown in the hippocampus of adult mice results in enhanced fear memory. Our results suggest that miR-153, and possibly other fear-induced miRNAs, act as components of a negative feedback loop that blocks neuronal hyperactivity at least partly through the inhibition of the vesicular transport pathway.

## Introduction

It is widely believed that the formation of stable memories involves changes in the strength of synaptic connections between neurons that are activated during learning (*Greer and Greenberg, 2008*; *Kandel, 2001*; *Lynch, 2004*). At the cellular level, sensory experience results in altered neurotransmitter release at the synapse, which triggers membrane depolarization and calcium influx into individual neurons. This action initiates a cascade of downstream events including the activation of protein kinases, redistribution of neurotransmitter receptors, and induction of changes in gene

expression, which together lead to stable changes in synaptic strength (*Flavell and Greenberg, 2008*; *Malinow and Malenka, 2002*; *Sutton and Schuman, 2006*). Although great progress has been made in describing how neuronal activation triggers downstream events, how the various induced pathways work together to coordinate changes at the synapse remains unknown.

Neuronal activation is associated with both transcriptional and post-transcriptional changes in gene expression that are required for modulation of synaptic plasticity. The role of de novo transcription and the functions of several families of transcription factors such as CREB, C/EBP$\beta$, Egr1, AP1, and Rel1 in synaptic plasticity and memory formation have been extensively studied (*Alberini and Kandel, 2015*). On the other hand, microRNAs (miRNAs) have emerged as a major class of regulators that act at the post-transcriptional level and control the expression of numerous target genes (*Bartel, 2009*). Hundreds of miRNAs have been identified in mammalian genomes (*Bartel, 2004*; *Lewis et al., 2003*), many of which are expressed in neurons (*Bartel, 2009*; *Friedman et al., 2009*; *Gaidatzis et al., 2007*; *Kozomara and Griffiths-Jones, 2014*; *Krek et al., 2005*; *Lewis et al., 2005*; *Lim et al., 2005*). Neuronal miRNAs play major roles in regulation of synaptic development and plasticity, and have been identified as components of regulatory pathways that modulate memory formation (*McNeill and Van Vactor, 2012*). For example, mouse miR-134 inhibits the expression of CREB, a key transcriptional regulator of genes involved in synaptic plasticity, and modulates synapse morphology by inhibiting the expression of LimK1 protein kinase (*Gao et al., 2010*; *Schratt et al., 2006*).

Recent studies have also identified activity-dependent changes in miRNA levels during memory formation. In one study, distinct populations of miRNAs that are up-regulated in the CA1 region of the hippocampus at early (1–3 hr) and late (24 hr) times after contextual fear-conditioning were identified and were proposed to positively regulate memory formation by increasing protein synthesis through de-repression of mTOR activity (*Kye et al., 2011*). In another study, miR-132 was found to be upregulated in the hippocampus upon induction of seizure or contextual fear-conditioning 45 min after training (*Nudelman et al., 2010*). However, the precise roles of activity-induced miRNAs in the hippocampus in vivo remain to be elucidated. Multiple hippocampal regions, in addition to the CA1, have been identified with the acquisition of different types of associative learning (*Milner et al., 1998*; *Rempel-Clower et al., 1996*). Studies using selective lesions within the hippocampus have demonstrated that all of the hippocampal regions participate in memory formation following a period of associative learning (*Jerman et al., 2006*; *Kubik et al., 2007*; *Lee and Kesner, 2004a*, *2004b*). The hippocampal regions with roles in the formation of memory associated with contextual fear conditioning are highly interconnected and may have distinct functions during memory formation. However, previous studies have not analyzed fear conditioning-induced changes in miRNA levels in the dentate gyrus region, which is also critical for the formation of memories associated with contextual fear conditioning.

We therefore aimed to identify fear conditioning-induced changes in the levels of miRNAs in the hippocampus, including the dentate gyrus region. Using a global expression profiling approach, we identified 21 miRNAs that are upregulated in the hippocampus of adult rats 24 hr after contextual fear conditioning. Four of these miRNAs are specifically induced as a result of associative learning and 12 are predicted to downregulate targets that are involved in vesicle exocytosis and synaptic plasticity processes. One of the miRNAs belonging to both above categories, miR-153, is transcriptionally induced, specifically in the dentate gyrus of the hippocampus, after contextual fear conditioning. We used a combination of in vitro and in vivo approaches to determine the role of miR-153 in both synaptic plasticity and long-term memory formation. Consistent with its induction pattern in vivo, in hippocampal brain slices miR-153 is specifically induced upon LTP induction in the dentate gyrus. Knockdown of miR-153 in the dentate gyrus results in more robust fear memory, suggesting that it plays a negative role in the regulation of synaptic strength. Consistent with this observation, overexpression of miR-153 in cultured hippocampal neurons reduces spine volume. Finally, consistent with the ability of miR-153 to suppress the expression of components of the SNARE-mediated vesicle exocytosis pathway, such as VAMP-2, and in support of a role for miR-153 as a negative regulator of synaptic plasticity, we discovered that overexpression of miR-153 suppresses the delivery of the AMPA receptors to the synapse in cultured neurons. Our findings suggest that miR-153 is a negative feedback regulator that is transcriptionally induced after contextual fear conditioning to downregulate changes that lead to increased synaptic strength. Furthermore, our bioinformatics analysis of the targets of the 21 fear-induced miRNAs, together with the validation of these targets for one

member of the group, suggests that down regulation of the vesicular transport pathway, which has previously been shown to be critical for synaptic remodeling, is a major component of the network of activity-induced changes in neurons.

## Results

### Identification of fear-induced miRNAs

To identify miRNAs that may function in formation of long-term memory, we performed expression profiling of miRNAs in the hippocampus of adult rats that were trained with the contextual fear conditioning paradigm (*Fanselow, 1980*). Adult female rats were trained with three foot shocks in a training chamber. We reasoned that miRNAs that are important for long-term memory may exhibit increased expression 24 hr after behavioral training had ceased. Although the distinct temporal phases or processes that take place during memory formation continue for several days, the transcription-dependent phase is relatively brief and is completed within 48 hr after training (*Alberini, 2011*). Animals were therefore sacrificed 24 hr after training and the hippocampus was dissected (*Figure 1A*). We isolated RNA from 18 hippocampi, prepared independent samples with pools of RNA from three trained or naïve animals, and using a miRNA microarray, identified 21 miRNAs that displayed at least 1.5-fold up-regulation in the hippocampus from trained rats compared to naïve rats in three biological replicates (*Figure 1B*; *Figure 1—figure supplement 1*).

To gain insight into the mechanisms by which this group of 21 miRNAs may regulate neuronal phenotype and synaptic transmission, we sought to identify their target mRNAs. Using a combination of the TargetScan, miRBase and microrna target prediction algorithms we identified over 3000 candidate genes (*Betel et al., 2008*; *Griffiths-Jones et al., 2008*; *Lewis et al., 2005*). We further reduced the number of candidates by requiring that the genes be expressed in the brain and possess binding sites for three or more of the 21 miRNAs. This strategy resulted in a list of 353 predicted targets (*Supplementary file 1A*). Network analysis conducted with these genes identified pathways essential to neuronal development, vesicle transport, long-term potentiation and synaptic contact (*Figure 1C*) (See Experimental Procedures). The vesicle exocytosis pathway emerged as one of the top three enriched networks, with 15 genes out of the 40 known components of the pathway identified as predicted targets that could be co-regulated by the 21 miRNAs. Further analysis of the 15 genes identified in this pathway revealed binding sites for 12 of the 21 miRNAs (*Supplementary file 1B*). These findings suggest that this group of miRNAs may be part of a regulatory network involved in suppressing vesicle exocytosis, a process that is required for neurotransmitter release, insertion of receptors at the synapse, and memory formation. MiR-153 emerged at the top of the list of 12 miRNAs responsible for regulating this group of genes (*Figure 1D,E*), with 12 potential targets in the vesicle exocytosis pathway. Interestingly 8 of these targets contained binding sites for three or more of the 21 fear-induced miRNAs based on our previous analysis (*Figure 1D,E*). The remaining 4 components of the vesicle exocytosis pathway had potential binding sites for either miR-153 alone, or together with one additional fear-induced miRNA (*Figure 1—figure supplement 2*). We therefore chose to investigate the possible role of miR-153 in memory formation and regulation of vesicle exocytosis in neurons.

### MiR-153 is a learning-induced miRNA

In order to form a long-term memory after contextual fear conditioning training, an animal must learn to associate a specific location (context) with a negative experience (foot shock). Several studies have indicated that contextual fear memory results from learning the context and shock alone, as well as the paired association (*Fanselow, 2000*) (*Frankland et al., 2004*). Therefore, distinct components of the animal's experience during contextual fear conditioning could differentially contribute to activation of the 21 fear-induced miRNAs. To determine the effect of the foot shock on miRNA expression, we analyze the expression of the fear-induced miRNAs in the hippocampus of adult rats that were trained with either an immediate shock test or the contextual fear conditioning paradigm (*Fanselow, 1980*) (*Figure 2A*). Adult female rats trained with the contextual fear conditioning paradigm were trained with three delayed foot shocks as described above. Rats in the immediate shock group were taken from their home cage, placed into the conditioning chamber, and immediately shocked with three foot shocks (*Figure 2A*). This behavioral paradigm does not allow the animal to

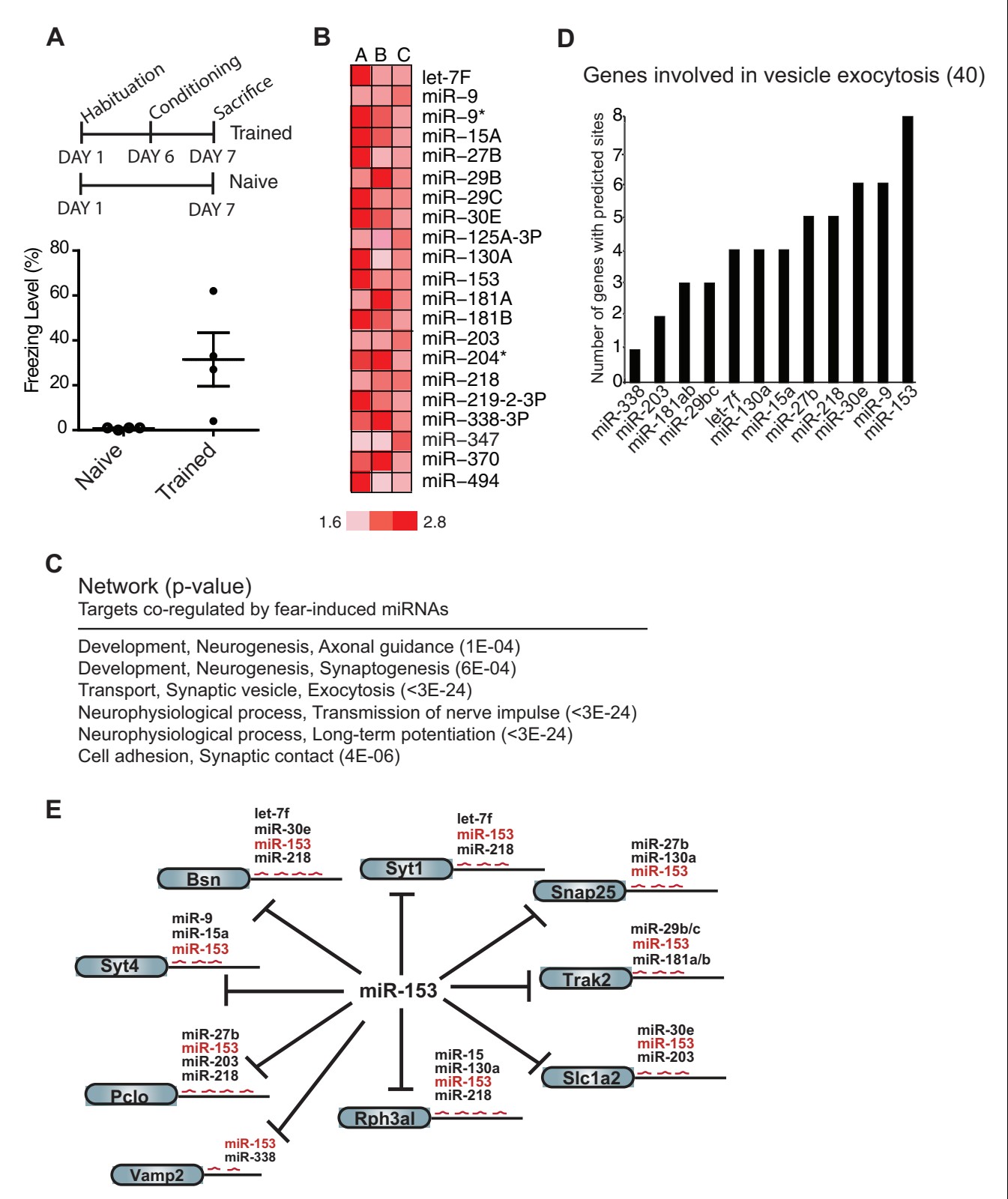

**Figure 1.** Expression profiling of miRNAs reveals 21 miRNAs that are induced in the hippocampus of adult rats 24 hr post-contextual fear conditioning. (**A**) Schematic representation of contextual fear conditioning paradigm. Rats were trained to associate an aversive unconditioned stimulus (foot shock) with the environment (context). Freezing behavior was examined 24 hr after contextual fear conditioning training for a control group of rats (n = 4 for each group, naïve and trained), a subset of rats for which tissue was not harvested. Error bars indicate SEM. P value from pairwise unpaired t-test is

*Figure 1 continued on next page*

*Figure 1 continued*

indicated with asterisks, *p<0.05. (B) MiRNAs that displayed at least a 1.5-fold increase in expression between trained and naïve rats in three different experiments. (C) Network analysis using MetaCore (Thompson Reuters) identifies pathways involved in neuronal development, vesicle exocytosis and synaptic plasticity that are co-regulated by three or more of the fear induced miRNAs identified in panel B. *P* values were calculated for each canonical signaling pathway as compared to the number of occurrences from random sets of brain-expressed genes (see Materials and methods for a detailed description of brain-expressed gene lists). All 6 of the pathways are statistically significant compared to random sets of brain-expressed genes, ***p<0.0001. (D) MiR-153 and miR-9 are the top two miRNAs co-regulating targets involved in the vesicle exocytosis pathway. (E) Eight predicted targets from the vesicle exocytosis pathway that may be co-regulated by miR-153 and at least two other fear-induced miRNAs. The potential targeting fear-miRNAs are indicated above each target.

The following figure supplements are available for figure 1:

**Figure supplement 1.** Identification of hippocampal fear-induced miRNAs.

**Figure supplement 2.** Additional targets shared between miR-153 and other fear-induced miRNAs.

form an associative memory between the context and the foot shock, as indicated by the absence of a freezing response in the immediate shock test group 24 hr after training (*Figure 2B*) (*Fanselow, 1986*). Animals were sacrificed 24 hr after training and the hippocampus was dissected. We isolated RNA from nine hippocampi for each naïve, immediate shock ('experience'), and delayed shock ('learning') groups, prepared independent samples with pools of RNA from three animals in each group (three biological replicates each containing three animals), and using RT-qPCR, analyzed hippocampal miRNA expression levels (*Figure 2C–E*; *Figure 2—figure supplement 1A–B*). This analysis revealed three distinct classes of fear-induced miRNAs. Class I miRNAs (miR-153, miR-181a, miR-204, miR-218) were induced only in the delayed shock group, indicating that their increased expression required associated learning (*Figure 2C*). Class II miRNAs (miR-27b, miR-219, miR-347) were induced in both the immediate and delayed shock groups, but displayed statistically significant higher expression levels in the delayed relative to the immediate shock group, suggesting that they were experience- and learning-induced (*Figure 2D*). Class III miRNAs (miR-9, miR-9*, miR-29b, and others) displayed increased expression in both the immediate and delayed shock groups, suggesting that they were experience-induced (*Figure 2E* and *Figure 2—figure supplement 1A*). The majority of the fear-induced miRNAs belong to class III miRNAs, which are likely induced after exposing rats to a novel context and the application of either immediate or delayed shock. Overall these findings reveal distinct classes of fear-induced hippocampal miRNAs, a subset of which is only induced as a result of associative learning and includes miR-153.

## MiR-153 levels increase specifically in the dentate gyrus during fear conditioning and LTP

To monitor the expression of fear-induced miRNAs in different hippocampal regions, we used quantitative real time PCR (RT-qPCR) on RNA preparations from the dentate gyrus and the CA1/CA3 regions of the hippocampus. We found ~10 fold higher levels of miR-153 expression in the dentate gyrus region of the hippocampus in trained compared to naïve rats but little or no change between trained and naïve animals in the CA1 and CA3 regions (*Figure 3A*). We also used RT-qPCR to validate the elevated expression in the hippocampus of trained rats for several of the other fear-induced miRNAs (*Figure 3—figure supplement 1A–E*). MiR-204, and to a lesser extent miR-9 and miR-125a were increased in the dentate gyrus; in contrast, miR-338–3p and miR-219–2 were not enriched in that same region, confirming that only specific miRNAs are enriched in the dentate gyrus 24 hr after behavioral training (*Figure 3—figure supplement 1A–E*).

In order to test whether increased miR-153 expression results from increased neuronal activity, we examined how miR-153 levels are affected after stimulation that results in long term potentiation (LTP). LTP in the hippocampus is a lasting form of synaptic plasticity that has been implicated in mammalian learning and memory (*Bliss and Collingridge, 1993*; *Martin et al., 2000*). One of the strongest correlations between the properties of behavioral memory formation and LTP is the evidence that each exhibits at least two mechanistically distinct phases of maintenance: an 'early' protein synthesis-independent phase that initiates synaptic changes and a transcription and protein

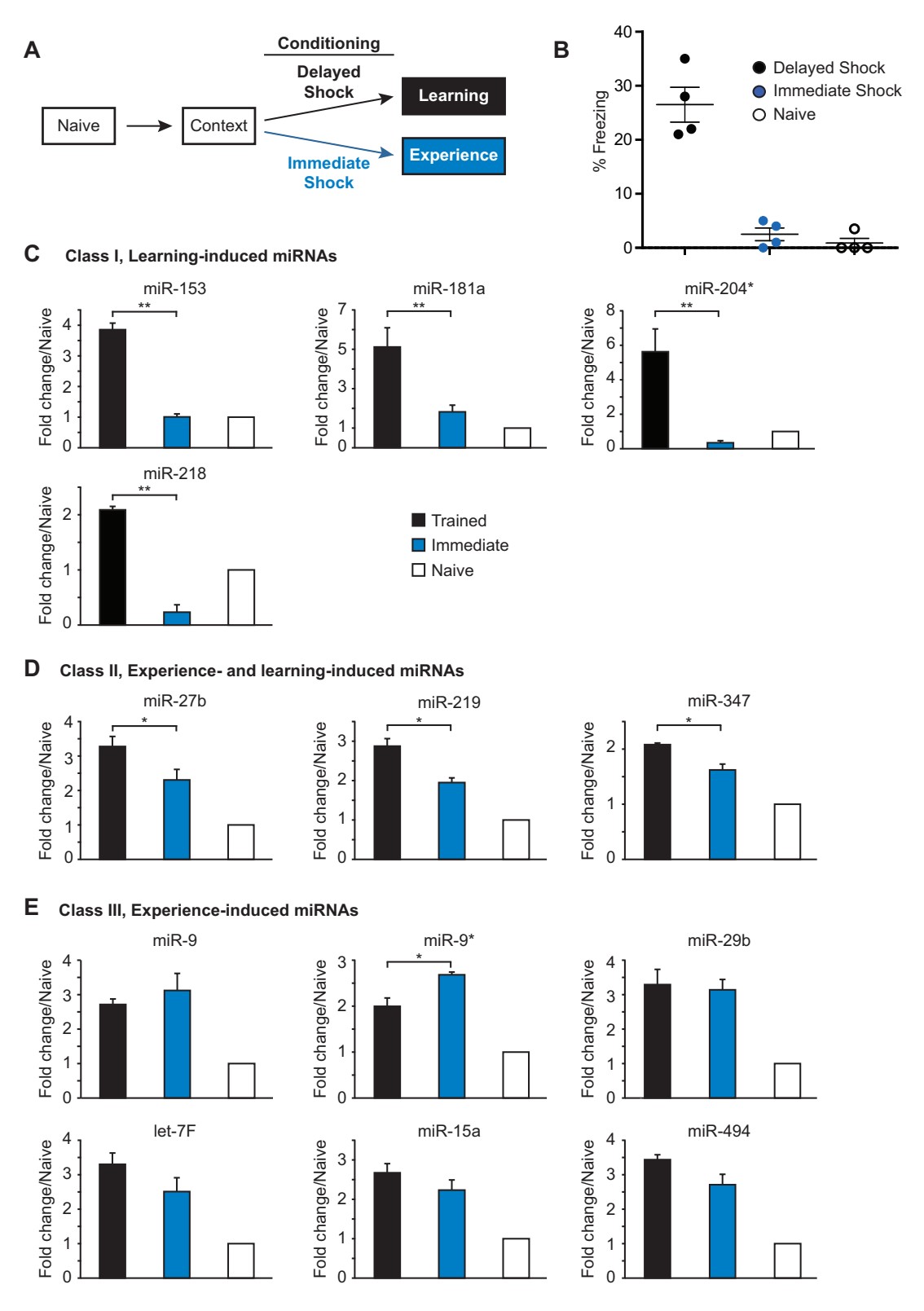

**Figure 2.** Identification of three classes of miRNA that are induced in the hippocampus with contextual fear conditioning. (**A**) Summary of experimental strategy. (**B**) Behavioral test of the naïve, immediate shock, and delayed shock animals. (**C–D**) RT-qPCR experiments showing changes in miRNA levels in the hippocampi of animals that were subjected to contextual fear conditioning with either the application of immediate shock after introduction to a novel context (immediate, blue bars) or delayed shock (trained, black bars) relative to animals that were only handled (naïve, white bars). (**C**) Class I

*Figure 2 continued on next page*

*Figure 2 continued*

miRNAs were specifically induced in the trained group. (**D**) Class II miRNAs were induced in both the immediate and trained groups with stronger induction in the trained group. The cases where the differences in miRNA levels between immediate and trained groups were statistically significant are indicated (\*p value<0.05). (**E**) Class III miRNAs were induced in both the immediate and trained hippocampi with no apparent increase in trained versus the immediate group. Each group contained nine animals, hippocampi from groups of three animals within each group were pooled for RNA isolation and RT-qPCR analysis.

The following figure supplement is available for figure 2:

**Figure supplement 1.** Class III miRNAs and control miRNAs.

synthesis dependent 'late' phase that contributes to activity-dependent structural changes (*Abraham and Otani, 1991*, *Abraham and Williams, 2003*; *Matthies et al., 1990*; *Ostroff et al., 2002*; *Raymond et al., 2000*). To assess whether the expression of miR-153 is induced during the maintenance of LTP, we examined perforant path-dentate gyrus LTP (PP-DG LTP) in acute hippocampal slices prepared from adult mice (*Figure 3B*). To facilitate further in vitro and genetic analysis of miR-153, we used mice as an experimental system for these and subsequent experiments. We note that miR-153 is highly conserved in vertebrates, displaying 100% sequence identity between rat, mouse, frog, and human (*Mandemakers et al., 2013*). Extracellular field potentials evoked by perforant path stimulation were recorded in the molecular layer of the dentate gyrus before and after high-frequency stimulation (HFS). LTP was quantified by examining HFS-induced changes in the slope of the field excitatory postsynaptic potential (EPSP) (*Figure 3C*). Three hours after LTP induction, which is around the beginning of the late phase LTP, the dentate gyrus, and CA1-CA3 regions were isolated and expression of miR-153 was quantified by RT-qPCR. Control regions were isolated from the remaining acute hippocampal slices following 3 hr of incubation in artificial cerebrospinal fluid in the absence of HFS, and expression of miR-153 was quantified by RT-qPCR. Consistent with the fear conditioning results, we observed elevated levels of miR-153 in the dentate gyrus, but not in the CA1-CA3 regions, after 3 hr of PP-DG LTP (*Figure 3D*), indicating that miR-153 expression is specifically induced in the dentate gyrus region during the late phase of LTP.

## miR-153 is transcriptionally induced from an alternative promoter

To better understand how fear-induced expression of miR-153 may be regulated, we surveyed for changes in chromatin modification patterns associated with active transcription after behavioral training. Histone H3 lysine 36 tri-methylation (H3K36me3) is associated with gene bodies of actively transcribed genes (*Li et al., 2007*). Of the miRNAs identified in the microarray panel, miR-153 was the only miRNA that showed increased enrichment of H3K36me3 across the coding sequence in trained compared to naïve rats (*Figure 4—figure supplement 1A,B*). In mammals, miR-153 is located within the 19th intron of Ptprn2, a host gene that is conserved among a wide range of phylogenetic taxa (*Xu et al., 2010*). Analysis of ENCODE H3K4me3 ChIP-seq data revealed two sites of enrichment within Ptprn2 (*ENCODE Project Consortium, 2012*). The first site of enrichment is located at the promoter of Ptprn2; the second site of enrichment is located within the gene body, 100 kilobases upstream of the miR-153 coding sequence. Since H3K4me3 is generally associated with promoter regions, the second site of enrichment for H3K4me3 within the Ptprn2 gene body may represent an alternative transcription start site that could independently drive the expression of a miR-153 precursor transcript. To test this hypothesis, we measured RNA expression changes at the Ptprn2 locus close to its canonical promoter and downstream of the second H3K4me3 peak in trained and naïve animals. We observed low levels of expression close to the Ptprn2 promoter, which were similar in trained and naïve animals (*Figure 4—figure supplement 1C*). In contrast, the expression of the transcript downstream of the internal H3K4me3 peak, as assayed for the exon closest to the miR-153 coding sequence, increased 1.4-fold in trained animals (*Figure 4—figure supplement 1C*), suggesting that miR-153 was probably generated by a separate transcriptional unit at the promoter found around the internal H3K4me3 region.

In order to gain insight into transcriptional regulation of miR-153, we first identified conditions in mature murine hippocampal cultures under which we could mimic the activity-induced overexpression observed in the hippocampus of trained rats following contextual fear conditioning. We performed a

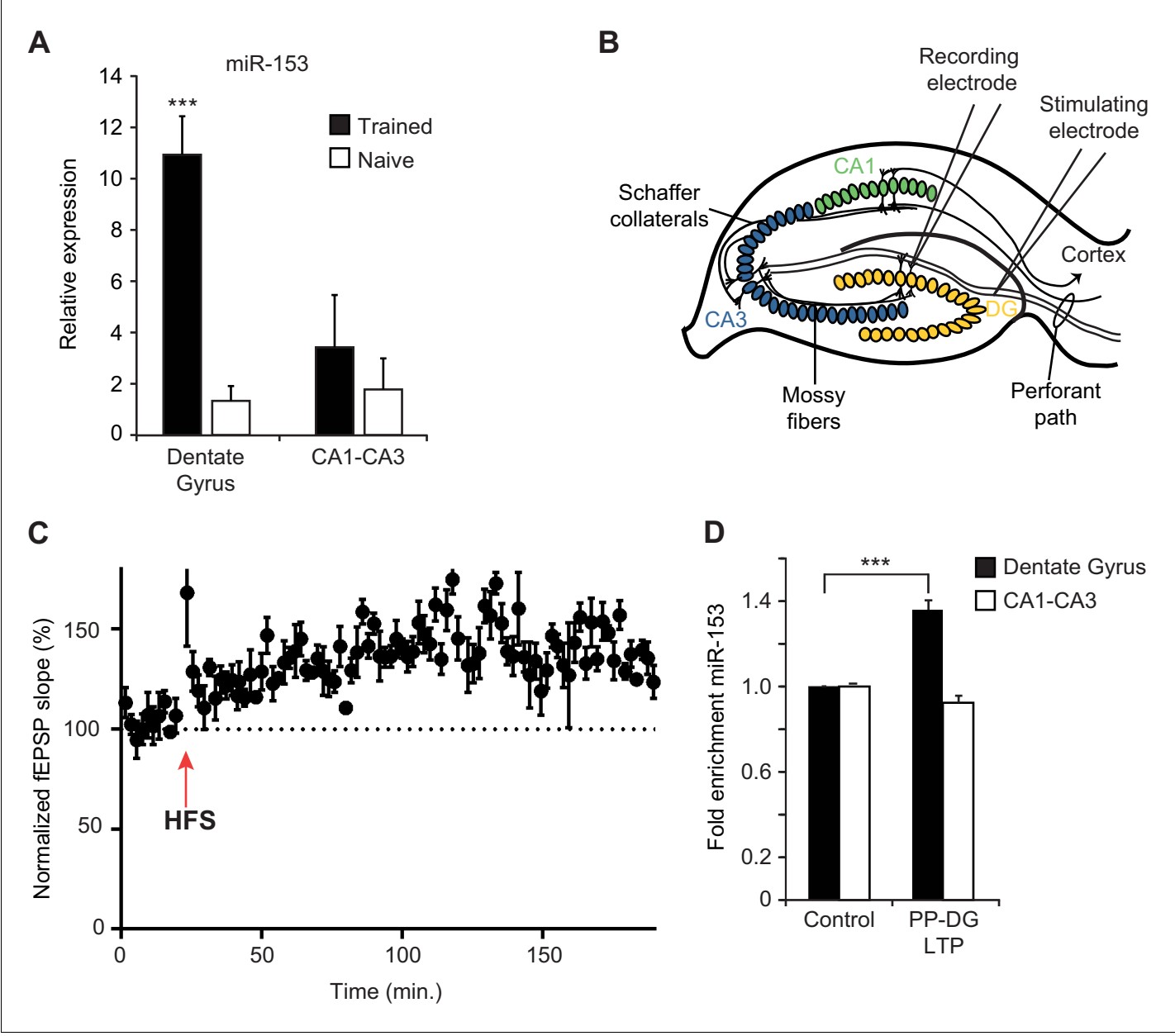

**Figure 3.** Expression of miR-153 is induced in the dentate gyrus by fear conditioning and LTP. (**A**) RT-qPCR analysis of RNA levels showing average expression profiles of miR-153 from the dentate gyrus and CA1-CA3 regions of hippocampus in naïve and trained rats. Expression levels were normalized to snoRNA-202 RNA levels. Error bars indicate standard deviation. ***p<0.001 (**B**) Schematic representation of a hippocampal slice showing stimulating and recording electrode sites. (**C**) Three-hour time course of perforant path-dentate gyrus (PP-DG) LTP in slices from wild-type mice (n = 5). A 20 min baseline was recorded, after which LTP was induced with four epochs of high frequency stimulation (labeled as HFS with a red arrow) applied 15 s apart. fEPSP slope was plotted demonstrating robust LTP even 3 hr after the induction (data points are averaged every 1.5 min). Each point represents mean ± SEM. (**D**) RT-qPCR analysis showing average expression relative to control for RNA isolated from the dentate gyrus and CA1-CA3 regions of hippocampal slices following 3 hr of PP-DG LTP. Expression levels were normalized to control RNA from the same region of the hippocampus. Error bars indicate standard error.

The following figure supplement is available for figure 3:

**Figure supplement 1.** Region-specific expression of hippocampal fear-induced miRNAs.

time course using mature murine hippocampal cultures, induced membrane depolarization by KCl, isolated small RNA at various time points, and quantified miR-153 expression by RT-qPCR. Expression of miR-153 was markedly increased 2–3 hr after membrane depolarization was induced with KCl (*Figure 4—figure supplement 1D*). On the other hand, an increase in the expression of the precursor form of miR-153 was observed 1 hr after membrane polarization, but was reduced to baseline expression levels 2–3 hr after KCl treatment, suggesting that a precursor RNA may be induced rapidly and then processed into mature miR-153 (*Figure 4—figure supplement 1E*).

We then performed chromatin immunoprecipitation (ChIP) assays to gain further insight into how miR-153 transcription is regulated during neuronal activation. We stimulated mature mouse hippocampal neurons with KCl for 3 hr and then assessed H3K4me3 levels at the promoter region of the host gene, *Ptprn2*, and the alternative promoter identified above, using tiling primers spanning those regions (*Figure 4A*). H3K4me3 levels were enriched at both promoter regions, however a much greater increase was observed at the internal promoter region compared to the *Ptprn2* promoter (3-fold versus 100-fold) (*Figure 4B–C*). Additionally, PolII Ser2 occupancy and H3K36me3 levels on the gene body of the miR-153 were significantly increased upon KCl stimulation (*Figure 4D*). Altogether, these data suggest that activation of hippocampal neurons leads to miR-153 transcriptional induction from an alternative promoter within the *Ptprn2* host gene.

We next used the TRANSFAC database to search for potential transcription factor consensus binding motifs across the entire alternative promoter sequence (*Figure 4A*). This search uncovered predicted binding sites for CBP/p300, CREB, C/EBPß, and ATF4 at the alternative promoter within the R3 and R5 sequences. We performed ChIP assays to examine possible activity-induced recruitment of each factor to DNA regions spanning H3K4me3 enrichment sites. We observed a dramatic increase in activity-dependent binding of each transcription factor following KCl stimulation with greater effects at the alternative promoter relative to the *Ptprn2* promoter (*Figure 4E,F*). These results suggest that the activity-dependent regulation of miR-153 may be mediated by a group of transcription factors, which have previously been implicated in regulation of activity-dependent transcription in neurons.

## Knockdown of miR-153 in the hippocampus enhances fear memory

We next examined the role of miR-153 in memory formation. To achieve this, we inhibited miR-153 activity in the dentate gyrus region of the hippocampus, via lentiviral mediated delivery of the miR-Zip-153 inhibiting vector with a GFP reporter (*Figure 5A*). Immunohistochemical analysis was performed to confirm proper targeting of the dentate gyrus by stereotaxic injection of the miRZip-153 and miRZip-scrambled (miRZip-scr) lentivirus, and to confirm that the observed phenotype was not due to damage of the dentate gyrus region as a result of the injections (*Figure 5—figure supplement 1A*). To further establish, at single-cell resolution, that the GFP[+] neurons were depleted of miR-153, we performed fluorescence-activated cell sorting (FACS) of dissociated hippocampal neurons from either miRZip-153-GFP or miRZip-scr-GFP mice and used them to prepare RNA for single-cell miR expression analysis. As shown in *Figure 5—figure supplement 1B*, the levels of miR-153 in GFP[+] neurons containing miRZip-153 were dramatically reduced compared to GFP[+] neurons containing the scrambled control miRZip. Inhibition of miR-153 function in the dentate gyrus resulted in a significant enhancement of long-term memory in the contextual fear-conditioning paradigm, whereas lentiviral-mediated delivery of a scrambled sequence did not affect fear memory (*Figure 5B*). In contrast, long-term memory in the cued fear-conditioning paradigm, which is predominantly amygdala-dependent (*Phillips and LeDoux, 1992*), was not affected by knockdown of miR-153 in the hippocampus (*Figure 5C*). Locomotion, anxiety-related and nociception behaviors were normal in miR-153 knockdown animals (*Figure 5—figure supplement 2A–M*) suggesting that the effect of miR-153 inhibition is hippocampal specific and is not due to differential pain sensitivity, motor coordination or anxiety levels in the injected groups of mice. These results demonstrate that miR-153 expression plays a specific role in attenuating contextual fear memory.

## miR-153 negatively regulates spine size

Several studies have suggested that changes in synaptic strength during memory formation and LTP induction correlate with corresponding changes in dendritic spine morphology. Dendritic spines are specialized actin-rich protrusions of the dendritic shaft. They are the major sites of excitatory

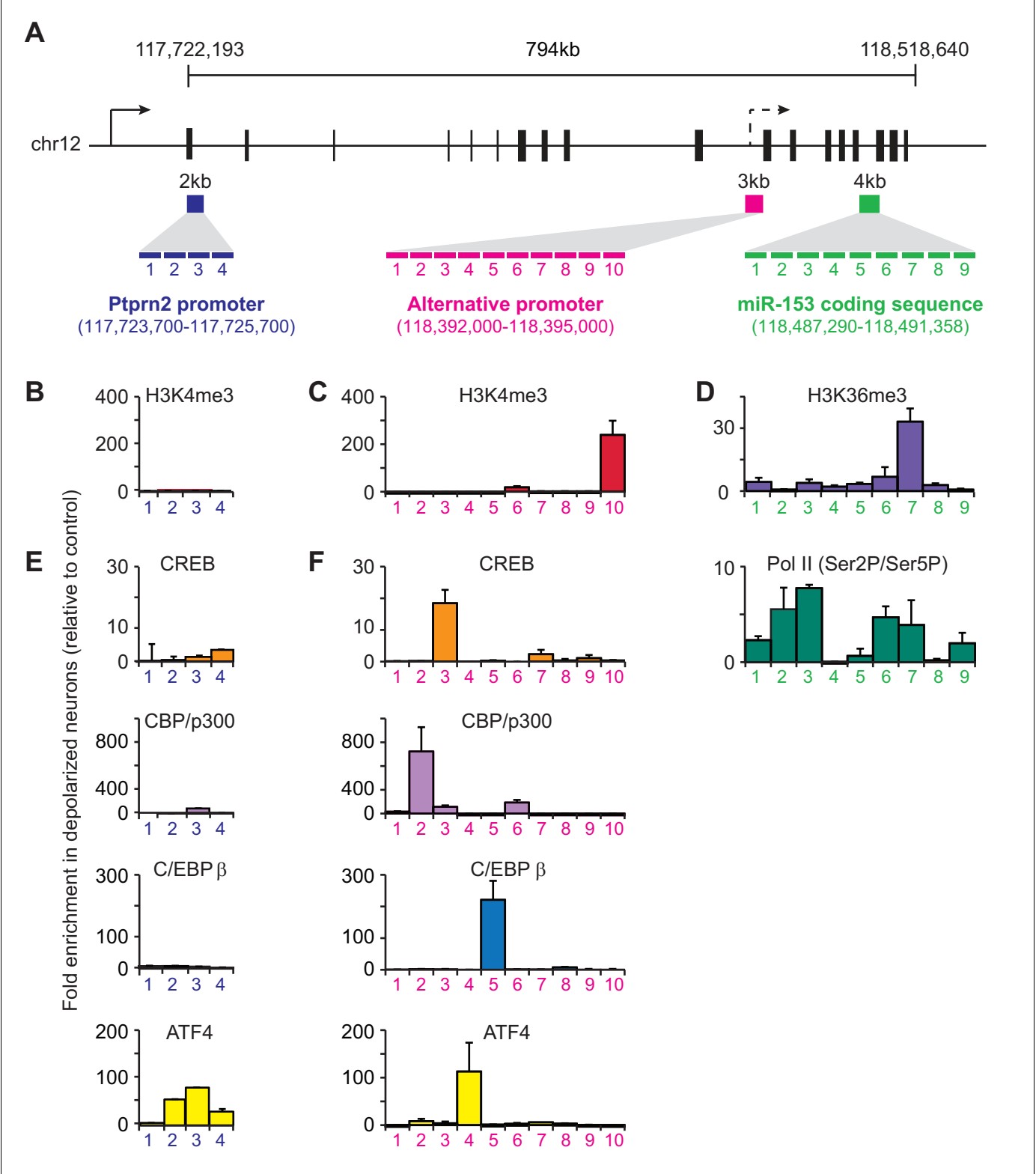

**Figure 4.** Transcriptional regulation of miR-153 expression may proceed through a cryptic promoter. (A) Schematic representation of Ptprn2 gene with regions identified for each tiling primer set used to map H3K4me3 for Ptprn2 (purple box, B), H3K4me3 for the cryptic promoter (magenta box, C), and H3K36me3 for the miR-153 coding sequence (green box, D). (B–C) ChIP-qPCR experiments showing changes in the association of histone H3K4me3 (red) with the (B) Ptprn2 promoter and (C) alternative cryptic promoter area. (D) ChIP-qPCR experiments showing changes in association of histone H3K36me3 (purple) and Pol II (green) across the miR-153 coding sequence. Tiling primer sets spanning a 2 kilobase range (1primer set/500 base pairs)

*Figure 4 continued on next page*

*Figure 4 continued*
were used to map the Ptprn2 promoter (**B**); a 3 kilobase range (1 primer set/300 base pairs) were used to map the putative cryptic promoter (**C**); and a 4 kilobase range (1 primer set/440 base pairs) were used to map the miR-153 coding sequence (**D**). (**E–F**) ChIP-qPCR experiments showing changes in association of CBP/p300 (purple), CREB (phosphorylated at Ser133, orange), C/EBPß (blue) and ATF4 (yellow) with the (**E**) Ptprn2 promoter and (**F**) alternative cryptic promoter area. All experiments were performed with chromatin isolated from mature mouse hippocampal neurons (14 DIV) depolarized continuously for 3 hr with 55 mM KCl relative to untreated hippocampal neurons. The experiments were performed in triplicate and the data are presented as mean ± standard deviation.

The following figure supplement is available for figure 4:

**Figure supplement 1.** H3K36 trimethylation occupancy is increased across the miR-153 coding sequence after contextual fear-conditioning and miR-153 is transcriptionally induced from an alternative promoter within Ptprn2.

synaptic connections and undergo morphological changes during development and in response to environmental stimuli (*Bourne and Harris, 2008*). MiR-153 overexpression after fear conditioning and LTP induction, as well as the more robust fear response of mice after the inhibition of its activity, prompted us to explore a possible role for miR-153 in the regulation of dendritic spine morphology. We performed time course RT-qPCR assays to detect the levels of miR-153 using primary hippocampal neurons at different days in vitro. We observed that miR-153 levels progressively increased and reached maximum levels after 10 days in vitro, when hippocampal neurons are differentiated and have made many synaptic connections, thus being more active (*Figure 6—figure supplement 1A*). Therefore, to test the effects of modulating miR-153 expression on dendritic spine geometry, we suppressed or enhanced miR-153 function in cultured mature hippocampal neurons by introducing either a miRZip-153 inhibiting vector to block the endogenous miR-153 activity or a miR-153 overexpressing vector to increase its levels (*Lovén et al., 2010*). The efficacy of these approaches was assessed using RT-qPCR to quantify miR-153 expression in neurons in which miR-153 was inhibited or overexpressed (*Figure 6—figure supplement 1B–C*). Analysis of the spine size in mature hippocampal neurons in which miR-153 was inhibited showed no significant change of the spine volume (*Figure 6A,B,E,F*, and *Figure 6—figure supplement 1D*). Although, we did not observe a change in average spine head width in miR-153 knockdown compared to control neurons (*Figure 6G*), analysis of the data as cumulative plots showed a significant increase in spine head width, most likely attributed to a group of spines of a particular size (*Figure 6—figure supplement 1E*). In contrast, miR-153 overexpression resulted in a significant decrease of the average spine volume, which was reduced to about 50% of the size of spines in the control scramble overexpressing hippocampal neurons (*Figure 6C–F*). Spine shrinkage in miR-153 overexpressing neurons was due to decreased spine head and neck widths (*Figure 6G–H* and *Figure 6—figure supplement 1E–F*). Neither overexpression of miR-153, nor its inhibition had any measurable or significant effect on the average spine density or length (*Figure 6I* and *Figure 6—figure supplement 1G–H*). The smaller effects observed in neurons where miR-153 is inhibited is likely due to the lower levels of miR-153 in un-stimulated neurons used in these experiments. Overall, these findings demonstrate that miR-153 activity modulates structural features of the synapse known to be associated with changes in plasticity.

## miR-153 down-regulates components of the vesicle exocytosis pathway and CBP/p300, an early-induced neuronal activity-dependent gene

Network analysis of the fear-induced miRs identified miR-153 as one of the top miRs predicted to regulate genes involved in SNARE-mediated vesicle exocytosis (*Figure 1D*). The SNARE complex is essential for vesicle exocytosis of several 'cargo' proteins that critically mediate spine growth during synaptic potentiation, AMPA receptor trafficking in the postsynaptic neuron, and neurotransmitter release from the presynaptic neuron during synaptic transmission (*Südhof, 2013*). Analysis of the full list of targets from the vesicle exocytosis pathway revealed a total of 12 predicted miR-153 targets (*Supplementary file 1*). Six predicted targets of miR-153 were selected from the vesicle exocytosis pathway for further analysis: *Snap25* and *Vamp2,* components of the SNARE complex, *Snca* and *Trak2* genes, which are important for vesicle trafficking, and *Bsn* and *Pclo* cytoskeletal associated proteins in the presynaptic area that are important for vesicle trafficking and exocytosis. Co-transfection of miR-153 into HEK-293T cells, a human cell line with undetectable levels of endogenous miR-

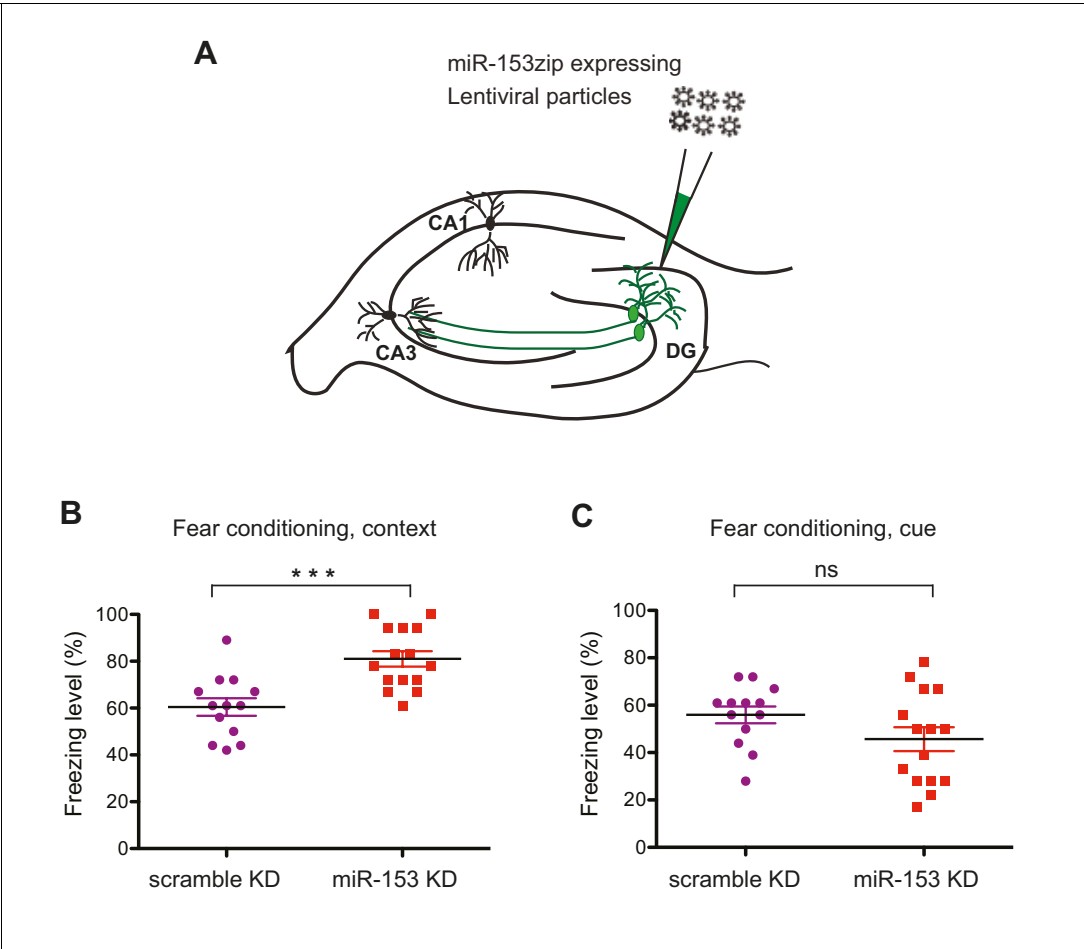

**Figure 5.** Knockdown of miR-153 enhances contextual fear-based memory. (**A**) Schematic representation of the DG region of the hippocampus that is injected by lentiviruses expressing miRZip-153 or miRZip-scramble. (**B**) miRZip-153 (KD) and miRZip-scramble (KD) injected mice were tested with a contextual fear conditioning task. Freezing behavior was examined 24 hr after contextual fear conditioning training. Contextual fear conditioning training was performed after injection of miRZip-153 (KD) or miRZip-scramble (KD) into the dentate gyrus region of the hippocampus. p=0.001 (**C**) miRZip-153 (KD) and miRZip-scramble (KD)-injected mice were tested with a cued fear conditioning task. Freezing behavior was examined 24 hr after cued fear conditioning training. Cued fear conditioning training was performed after injection of miRZip-153 (KD) or miRZip-scramble (KD) into the dentate gyrus region of the hippocampus.

The following figure supplements are available for figure 5:

**Figure supplement 1.** miR-153 (KD)-GFP and scrambled-GFP in the dentate gyrus of C57BL/6 mice.

**Figure supplement 2.** Behavioral characterization of miR-153 (KD)-GFP and scrambled-GFP injected mice.

153 (*Doxakis, 2010*), with the *Renilla* luciferase reporter fused to the 3'UTR of *Snap25*, *Vamp2*, *Snca*, *Trak2*, *Bsn* and *Pclo* conferred >40% decrease in luciferase activity, indicating that each was a direct target of miR-153 (*Figure 7A*, purple bars). Mutation of the miR-153 seed region in the 3'-UTR region of the above genes abrogated its inhibition by miR-153 expression, indicating that miR-153 directly targets each mRNA for silencing (*Figure 7A*, white bars). Next, we overexpressed exogenous miR-153 using lentiviral-mediated delivery in cultured mature hippocampal neurons to determine how miR-153 levels affect the expression of the above target genes. As shown in *Figure 7B*, miR-153 overexpression resulted in a significant decrease in mRNA levels for each of the above targets except *Trak2*. MiR-153 therefore downregulates components of the vesicle transport pathway in both HEK293 and cultured mature hippocampal neurons.

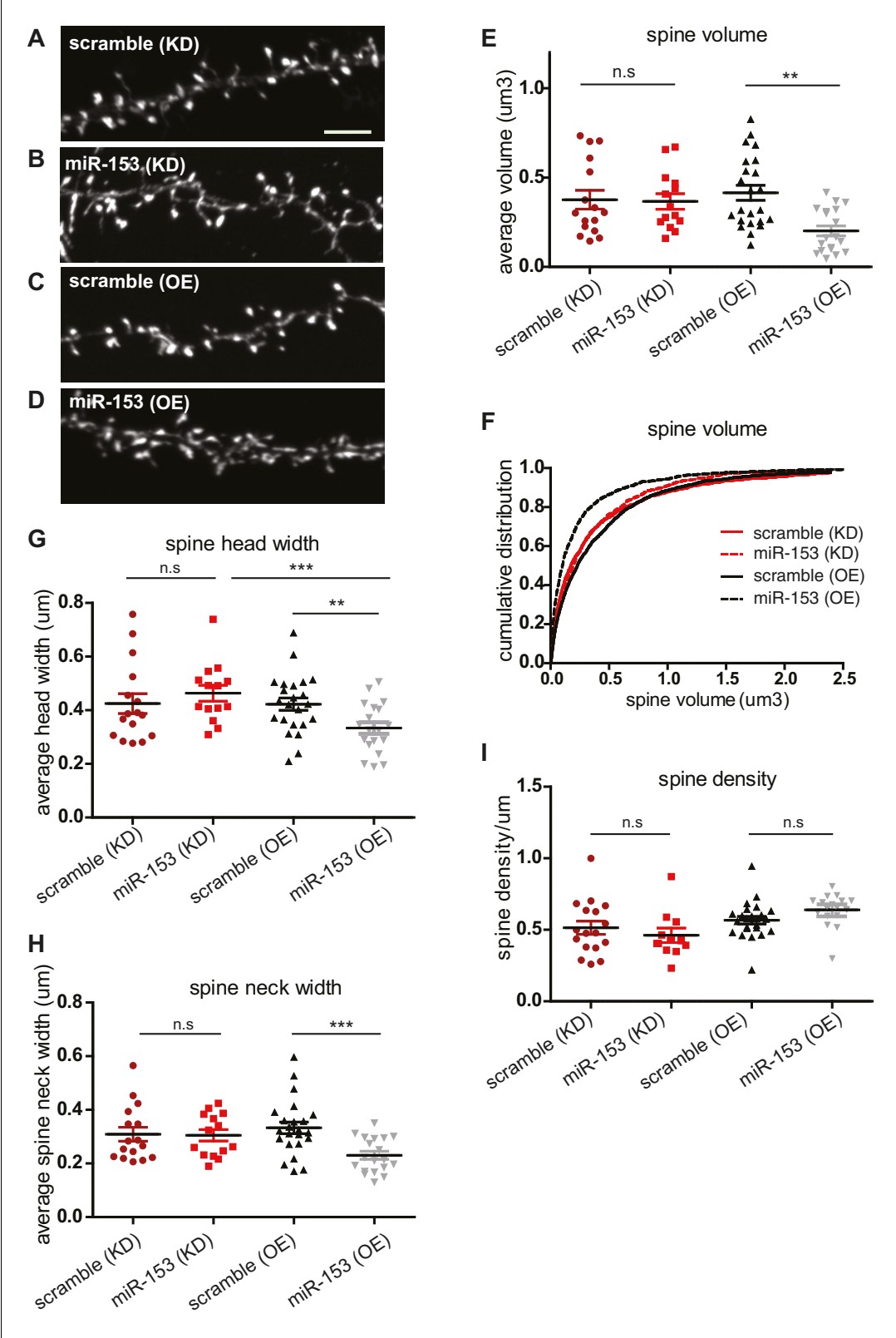

**Figure 6.** Overexpression of miR-153 decreases dendritic spine volume in hippocampal neurons. (A–D) Lifeact-mRuby images, used to visualize spines, of dendritic regions from representative neurons (DIV18) transfected with scramble (KD) control vector (A), miR-153 (KD) inhibiting vector (B), scramble (OE) control vector (C), miR-153 (OE) overexpressing vector (D). Scale bar, 5 μm. KD represents knock-down and OE represents overexpression. (E) Average volume of spines (n > 800 spines) from neurons (n = 14–23) transfected as in (A–D). The means for these parameters were calculated for each
*Figure 6 continued on next page*

*Figure 6 continued*

neuron separately and then the averages from all neurons together were plotted as average numbers for each condition. Error bars indicate SEM. *P* values of pairwise unpaired t-tests are indicated with asterisks scramble (KD) vs miR-153 (KD) (p=0.84) and scramble (OE) vs miR-153 (OE) (p=0.003). *p<0.05, **p<0.01 (F) Cumulative distributions of spine volume were plotted for each group of hippocampal neurons (DIV18) transfected as above. At least 800 spines were measured from two independent experiments and three coverslips each experiment per condition. Spine volume was decreased in miR-153 overexpressing neurons (p<0.0001, D = 0.21), but not changed in neurons with miR-153 inhibited (p=0.37, D = 0.041). (G) Average spine head width calculated and represented as in (E). Error bars indicate SEM. *P* values of pairwise unpaired t-tests are indicated with asterisks; scramble (KD) vs miR-153 (KD) (p=0.43), scramble (OE) vs miR-153 (OE) (p=0.008) and miR-153 (KD) vs miR-153 (OE) (p=0.001). *p<0.05, **p<0.01, ***p<0.001 (H) Average spine neck width calculated and represented as in (E). Error bars indicate SEM. *P* values of pairwise unpaired t-tests are indicated with asterisks scramble (KD) vs miR-153 (KD) (p=0.9) and scramble (OE) vs miR-153 (OE) (p=0.0006). ***p<0.001 (I) Average spine density of hippocampal neurons calculated as in (E). Error bars indicate SEM. *P* values of pairwise unpaired t-tests are indicated with asterisks. n.s., not significant.

The following figure supplement is available for figure 6:

**Figure supplement 1.** miR-153 negatively regulates dendritic spine size.

CREB, C/EBPß, and CBP/p300 are part of a group of early-induced neuronal activity-dependent genes. These immediate early genes are considered critical regulators of a gene expression program that is induced to promote strengthening of synaptic connections. Analysis of the list of predicted targets for miR-153 revealed CBP/p300 as a candidate mRNA (*Figure 7—figure supplement 1A*). To test whether CBP/p300 was a direct target of miR-153, we cloned its 3'-UTRs downstream of the *Renilla* luciferase coding sequence and transfected each plasmid together with a vector overexpressing pri-miR-153 into HEK293T cells, as described above for vesicle exocytosis targets. Co-transfection of miR-153 with the *Renilla* luciferase reporter fused to the 3'UTR of *CBP/p300* conferred a >70% decrease in luciferase activity (*Figure 7—figure supplement 1B*). Mutation of the miR-153 seed sequence in the 3'UTR of *CBP/p300* abolished this decrease, indicating that *CBP/p300* was a direct target of miR-153 (*Figure 7—figure supplement 1B*). These findings suggest that miR-153 downregulates expression of CBP/p300 and may provide feedback control to return this transcription factor to its basal level of expression (*Figure 7—figure supplement 1C*).

To determine whether the expression levels for vesicle exocytosis target genes were repressed after fear conditioning, we performed gene expression analysis on hippocampal RNA isolated from naïve and trained rats. We observed a reduction in *Pclo, Snca, Snap25, Trak2* and *Vamp2*, but not *Bsn*, expression levels after fear conditioning (*Figure 8A–F*). To further evaluate these targets in vivo in hippocampal GFP$^+$ neurons that were depleted of miR-153, we performed fluorescence-activated cell sorting (FACS) of dissociated hippocampal neurons from either miRZip-153-GFP or miRZip-scr-GFP injected mice and used them to prepare RNA for gene expression analysis (*Figure 8—figure supplement 1A–C*). We observed a 2- to 3-fold increase in *Pclo* and *Vamp2* expression levels in GFP$^+$ neurons after knockdown of miR-153 compared to both GFP$^-$ and scramble control neurons (*Figure 8G,J*). Expression levels were unchanged for *Snap25* and *Snca* and were not detectable for *Bsn* and *trak2* in GFP$^+$ neurons after knockdown of miR-153 compared to both GFP$^-$ neurons and scramble control neurons (*Figure 8H,I*). Taken together, these observations suggest that miR-153 may downregulate components of both the presynaptic (*Pclo* and *Vamp2*) and postsynaptic (*Vamp2*) vesicle exocytosis pathways.

## MiR-153 regulates AMPA receptor exocytosis and glutamate release

In order to determine whether miR-153-mediated regulation of vesicular transport components affects the presynaptic terminus we employed two strategies. We monitored functional nerve termini using the dye FM4–64 in the presence of KCl-mediated neuronal excitation. We observed that lentiviral-mediated inhibition of miR-153 in primary hippocampal neurons resulted in faster depletion of the FM4–64 dye compared to control neurons, suggesting a higher rate of vesicle exocytosis promoted by KCl-depolarization in the absence of miR-153 (*Figure 9A*). Conversely, lentiviral-mediated overexpression of miR-153 in hippocampal neurons resulted in a slower rate of FM4–64 dye depletion, indicating that the rate of vesicle exocytosis was decreased as a result of miR-153 overexpressed (*Figure 9B*).

Alternatively, we examined the effect of miR-153 depletion on glutamate secretion using cultured primary hippocampal neurons and a hippocampal cell line (H19-7). We transduced cultured

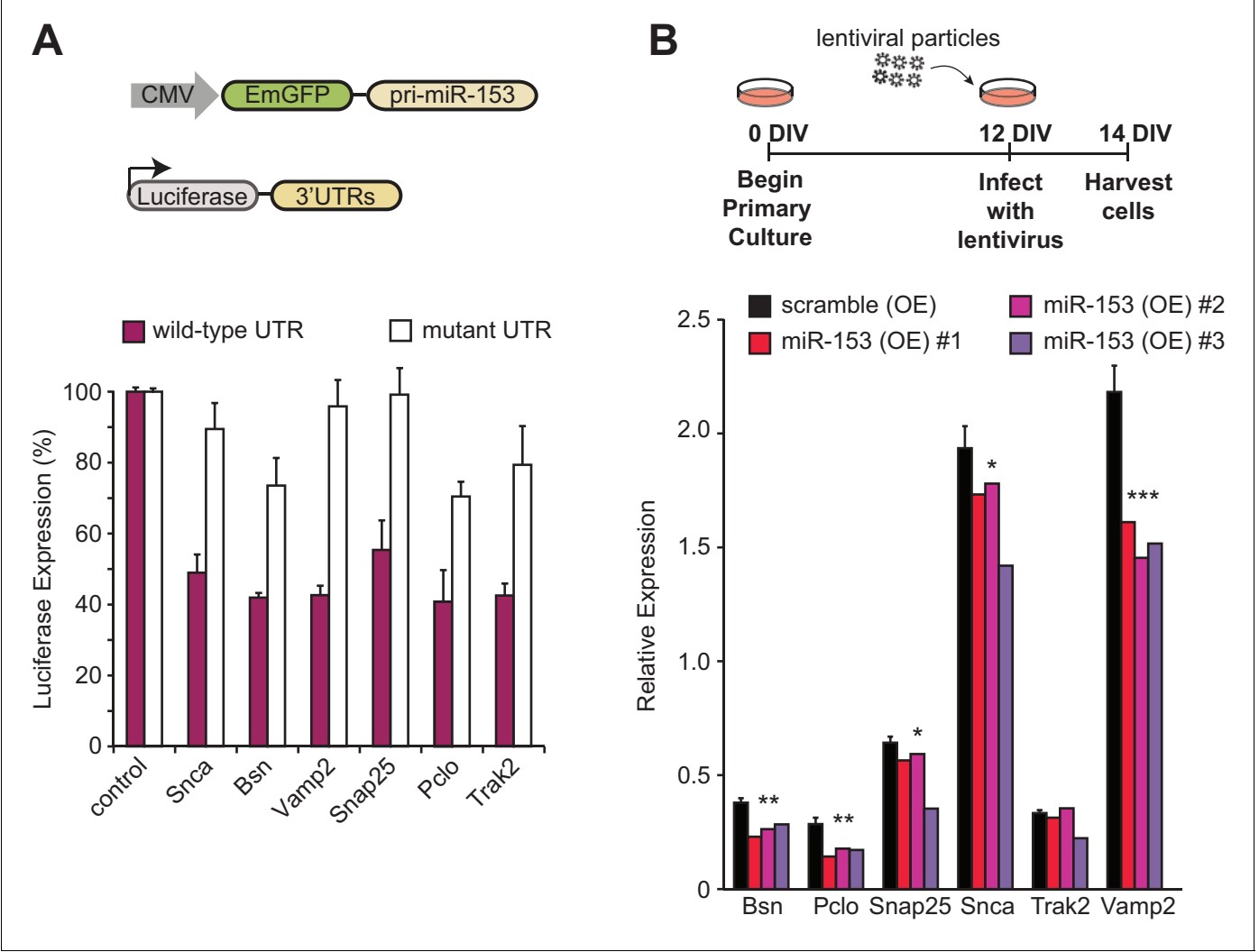

**Figure 7.** miR-153 regulates the expression of targets involved in the vesicle exocytosis pathway. (**A**) Cells with luciferase reporter constructs containing wild-type (purple bars) or mutant (white bars) *Bsn, Pclo, Snap25, Snca, Trak2,* or *Vamp2* mouse 3'-UTR region were co-transfected with the miR-153 expression plasmid. A miR-153 scrambled expression plasmid served as a control (leftmost bars). 3'-UTR mutations were in the seed sequence for the miR-153 binding site for each gene. HEK293T cells were co-transfected with both the reporter gene and miRNA expression vectors, and luciferase activity was measured 48 hr later. The experiments were performed in triplicate and error bars indicate standard deviation. (**B**) RT-qPCR analysis of RNA levels for genes (*Bsn, Pclo, Snap25, Snca, Trak2,* and *Vamp2*) from miRZip-153 (OE) infected 14 DIV hippocampal neurons (red, magenta and purple bars), relative to neurons infected with miRZip-scramble (OE), a control scrambled miR lentivirus (black bar). Each experiment was performed in triplicate. Error bars indicate standard deviation. *p<0.05, **p<0.01, ***p<0.001.

The following figure supplement is available for figure 7:

**Figure supplement 1.** CBP/p300 is a target of miR-153.

hippocampal neurons with miRZip-153 and miRZip-scr lentivirus and quantified the release of [H$^3$] glutamate after stimulation with 55 mM KCl. Knockdown of miR-153 in cultured hippocampal neurons increased the release of [H$^3$]-glutamate compared to the scramble control (*Figure 9C*). Similar results were obtained with stably transfected and differentiated H19-7 cells, which are competent for glutamate release (*Akchiche et al., 2010*) and in which miR-153 is also activity-induced (*Figure 9—figure supplement 1A* and *Figure 9—figure supplement 1B–C*). Furthermore, cultured hippocampal neurons transduced with miR-153 overexpression lentivirus and stimulated with 55 mM KCl secreted about 49% less [H$^3$]-glutamate than scramble control transfected cells (*Figure 9D*). The

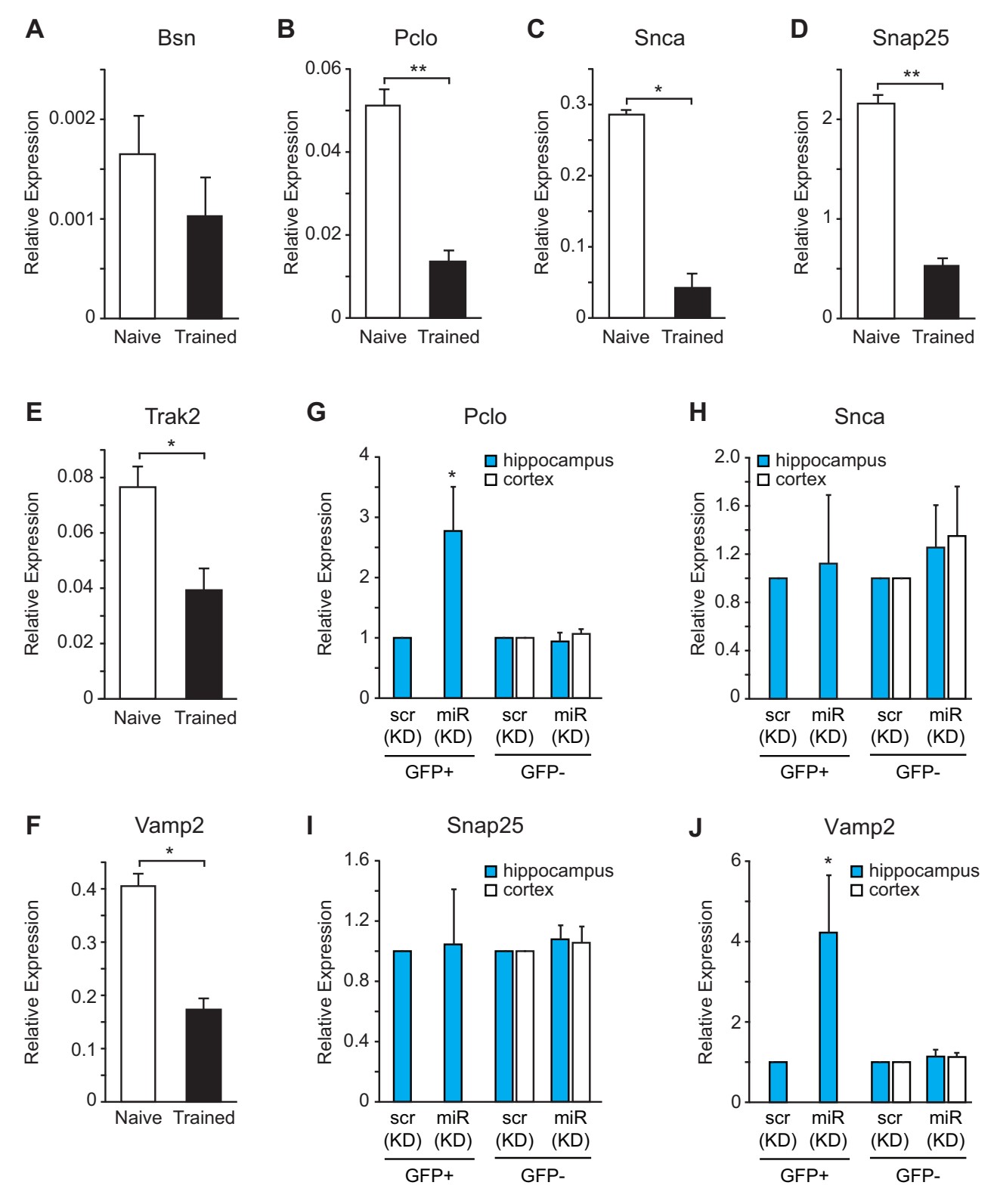

**Figure 8.** miR-153 regulates targets involved in the vesicle exocytosis pathway that are suppressed after fear conditioning. (A–F) RT-qPCR analysis of RNA levels for vesicle exocytosis genes *Bsn* (A), *Pclo* (B), *Snca* (C), *Snap25* (D), *Trak2* (E), and *Vamp2* (F) from hippocampus of naïve and trained rats. (G–J) RT-qPCR analysis of RNA levels for vesicle exocytosis genes *Pclo* (G), *Snca* (H), *Snap25* (I), and *Vamp2* (J) from hippocampus and cortex tissue isolated from miRZip-153 (KD) and miRZip-scr (KD) injected mice. Transcript levels are reported for FACS sorted GFP+ or GFP− neurons from miRZip-

*Figure 8 continued on next page*

*Figure 8 continued*

153 (KD) tissues relative to FACS sorted GFP$^+$ or GFP$^-$ neurons from miRZip-scr (KD), a control scrambled miR lentivirus. Each experiment was performed in triplicate. Error bars indicate standard deviation. *p<0.05.

The following figure supplement is available for figure 8:

**Figure supplement 1.** Fluorescence-activated cell sorting (FACS) plots of hippocampal neurons isolated from wild-type mice transduced in the dentate gyrus with miRZip lentiviruses.

change in [H$^3$]-glutamate release depended on KCl concentrations required to depolarize neurons as neither knockdown or overexpression of miR-153 under low KCl concentrations induced a change in [H$^3$]-glutamate levels (*Figure 9C–D* and *Figure 9—figure supplement 1D–E*). These findings suggest that miR-153 suppresses depolarization-induced glutamate secretion.

Previous studies have identified a unique SNARE complex, including VAMP2 with roles in the exocytosis of AMPA receptor subunits at the post-synaptic area (*Jurado et al., 2013*). In order to assess whether miR-153 has a role in regulating AMPA receptor exocytosis, we monitored the exocytosis of GluA1 subunit using fluorescence recovery after photobleaching (FRAP). We used differentiated neuroblastoma cells (N2A) that do not express miR-153 in order to examine the effect of ectopic miR-153 expression on AMPA receptor exocytosis. Differentiated N2A cells were transfected with a miR-153 expression vector or a scramble expression vector carrying an mCherry cassette to visualize cells expressing miR-153 along with a construct expressing the GluA1 subunit of AMPA receptor tagged with a pH-sensitive form of EGFP (Super Ecliptic pHluorin, SEP) that displays fluorescence when it is present mainly on the cell surface (*Figure 9—figure supplement 2A–D*). This strategy provides the opportunity to capture exocytosis of GluA1 receptor when it moves to the cell membrane from intracellular compartments. Three days after transfection, differentiated N2A cells were photobleached to eliminate SEP fluorescence and images were acquired every 10 min for 150 min (*Figure 9—figure supplement 2A*). After photobleaching, SEP fluorescence in control cells gradually recovered by the end of the time course (*Figure 9—figure supplement 3A,C*). In contrast, in cells expressing miR-153, SEP fluorescence did not recover to the same levels (*Figure 9—figure supplement 3B–C*). The total levels of GluA1 were comparable between scramble expressing cells and miR-153 expressing cells as observed with western blotting, indicating that miR-153 expression did not affect GluA1 protein levels (*Figure 9—figure supplement 2E*). These findings suggest that miR-153 inhibits the transport of the GluA1 AMPA receptors to the cell surface.

## Discussion

In this study, we have identified a group of hippocampal miRNAs that are upregulated following contextual fear conditioning. Our analysis of their predicted targets suggests that components of the vesicular transport and synaptogenesis pathways are major substrates for regulation by these miRNAs. Further analysis of, miR-153, which is specifically induced as a result of associative learning, revealed that it inhibits the expression of several components of the vesicular transport machinery, regulates neuronal features, such as spine morphology, which correlate with LTP induction and learning, inhibits the transport of GluA1 AMPA receptors to the surface of neurons, and inhibits glutamate release/uptake. As would be expected from a functionally important learning-associated factor, miR-153 expression is specifically induced in dentate gyrus granule cells during LTP in hippocampal slices, and its knockdown in the dentate gyrus region of the hippocampus in adult mice results in enhanced fear memory. Together, our results suggest that microRNAs, such as miR-153, are experience- or learning-induced to act as negative feedback regulators of pathways associated with synaptic plasticity, such as the vesicle transport pathway, and may serve to attenuate changes associated with increased synaptic strength, such as the delivery of AMPA receptors to the synapse and neurotransmitter release from the presynaptic area (*Figure 9E*).

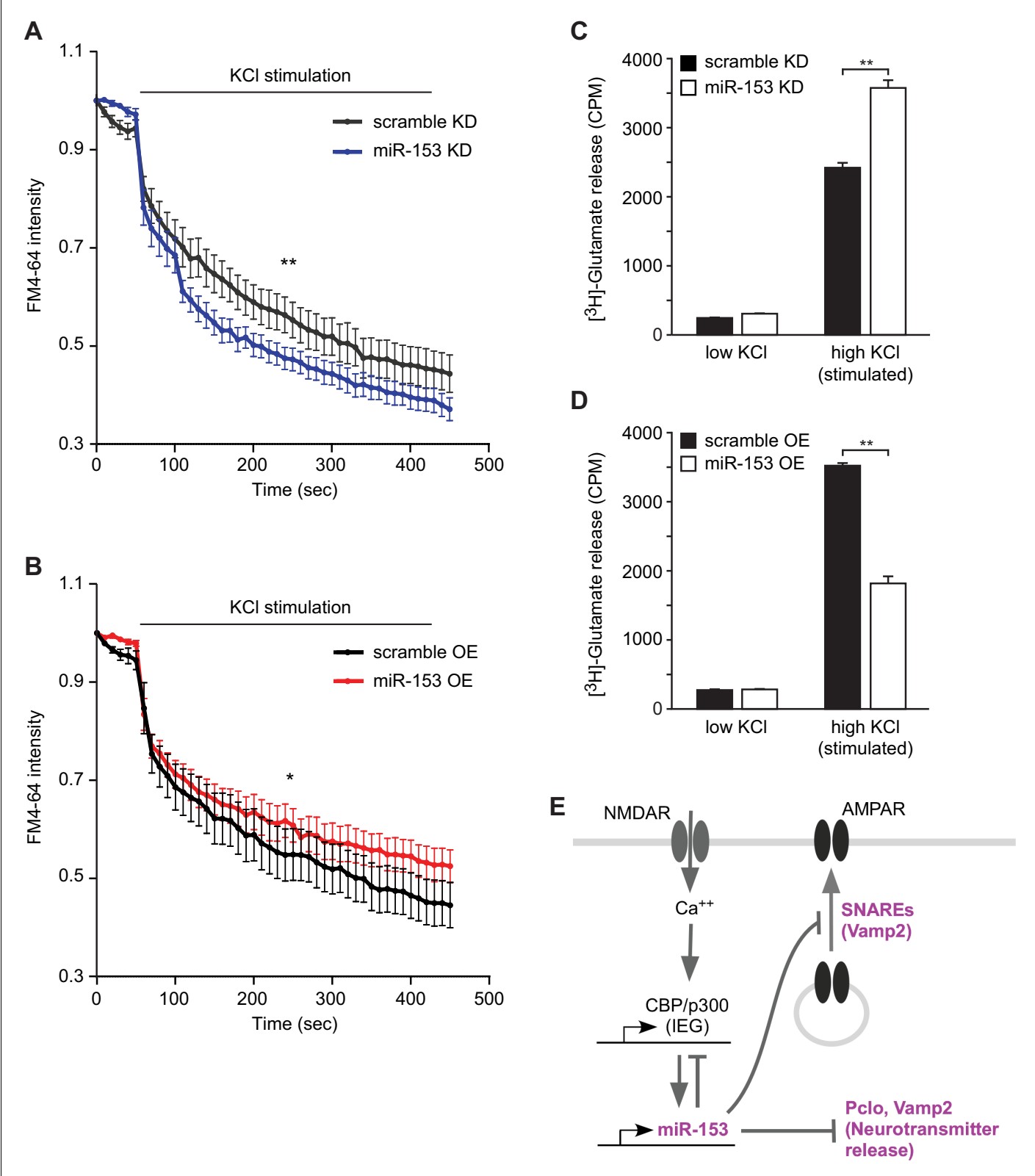

**Figure 9.** miR-153 regulates vesicle exocytosis and glutamate neurotransmitter release. (A–B) Analysis of vesicle exocytosis by FM4–64 imaging in primary neurons. (A) Fold fluorescence depletion of FM4-–64 dye of scramble (KD) and miR-153 (KD) groups of (n = 10–12) neurons per group, **p=0.008, two-tailed Mann Whitney U test. (B) Fold fluorescence depletion of FM4-–64 dye of scramble (OE) and miR-153 (OE) groups of (n = 10–12) neurons per group, *p=0.014, two-tailed Mann Whitney U test. (C–D) [H$^3$]-glutamate release in primary neurons. (C) [H$^3$]-glutamate release as

*Figure 9 continued on next page*

Figure 9 continued

determined by measuring the radioactivity content in neurons transduced with miRZip-153 knockdown (KD) lentivirus (white bars) as compared to primary neurons transduced with miRZip-scr (KD) lentivirus (black bars) after depolarization with 55 mM KCl (high KCl). (D) [H³]-glutamate release as determined by measuring radioactivity content in neurons transduced with miR-153 overexpression (OE) lentivirus (white bars) as compared to primary neurons transduced with scramble overexpression (OE) lentivirus (black bars) after depolarization with 55 mM KCl (high KCl). Each experiment was performed in triplicate. Error bars indicate standard deviation. *p<0.05. (E) Schematic summary illustrating the role of miR-153 as a negative feedback regulator of the pathways that mediate changes in synaptic strength and neurotransmitter release.

The following figure supplements are available for figure 9:

**Figure supplement 1.** Neurotransmitter uptake and release measurements in H19-7 hippocampal neuronal cells.

**Figure supplement 2.** Fluorescence recovery after photobleaching (FRAP) of SEP-GluA1 AMPA receptors in neuroblastoma N2A cells.

**Figure supplement 3.** miR-153 regulates AMPAR transport in neuroblastoma N2A cells.

## The vesicular transport pathway as a primary target of experience- and learning-induced miRNAs

The functional adaptations that occur during LTP and memory formation often rely on rapid changes in the composition of postsynaptic membranes. In the mammalian brain, synaptic plasticity requires regulated trafficking of AMPA receptors at excitatory synapses, which mediates learning-induced changes in the number and stoichiometry of postsynaptic neurotransmitter receptors (*Matsuo et al., 2008*; *Whitlock et al., 2006*). More specifically, NMDA receptor-triggered LTP involves exocytosis of GluA1 AMPA receptor subunits and an increase in their density at the postsynaptic membrane (*Hayashi and Huganir, 2004*; *Lu et al., 2001*; *Park et al., 2004*; *Passafaro et al., 2001*; *Pickard et al., 2001*; *Shi et al., 1999*). Receptor exocytosis is mediated by the soluble NSF-attachment protein receptor (SNARE) pathway, which attaches intracellular vesicles to their target membranes and drives membrane fusion. In this study, we show that VAMP2 and SNAP25 SNAREs are post-transcriptionally regulated by somatodendritically localized miR-153. The negative effect of miR-153 on VAMP2 and SNAP25 expression as well as on the delivery of the GluA1 receptors to the cell surface, together with the well-established role of GluA1 receptor transport in LTP (*Figure 7* and *Figure 8I,J*; *Shepherd and Huganir, 2007*), provides an explanation for enhancement of fear memory following the knockdown of miR-153 in the dentate gyrus region of the hippocampus (*Figure 5B*). Furthermore, miR-153 expression is induced in the dentate gyrus during LTP in acute hippocampal slices via stimulation of the perforant path-dentate granule cell neuronal pathway (*Figure 3B–D*). This is the physiological pathway for input of fear-associated sensory information into the dentate gyrus. Together the results support the idea that activation of a learning-associated neuronal pathway induces miR-153 expression in the brain to modulate memory formation. Finally, our in silico analysis revealed an additional 11 fear-induced miRNAs that may potentially target vesicle exocytosis and participate in AMPA receptor trafficking at the postsynaptic area. Therefore, AMPA receptor trafficking is likely to be regulated by the coordinated action of a large number of fear-induced miRNAs.

The importance of the vesicle exocytosis pathway in learning and memory has been underscored by several previous studies. ShRNA-mediated knock-down of several SNARE proteins in combination with high-resolution live cell imaging has demonstrated a role for a unique SNARE-dependent fusion machinery in exocytosis of AMPA and NMDA receptors at the postsynaptic area (*Jurado et al., 2013*). The R-SNARE protein VAMP2 and Q-SNARE protein SNAP25 are essential components of the postsynaptic vesicle fusion machinery that is required for neurotransmitter receptor trafficking during basal- and LTP-induced neuronal activity, suggesting that these SNAREs are important for both constitutive and regulated exocytosis (*Jurado et al., 2013*). VAMP2 contributes to constitutive and regulated postsynaptic AMPA receptor trafficking, while SNAP25 plays a role in the constitutive postsynaptic trafficking of NMDA receptors (*Jurado et al., 2013*). In our studies, miR-153 overexpression in cultured hippocampal neurons resulted in decreased expression of both VAMP2 and SNAP25, indicating that both SNAREs could be regulated by miR-153. However, in vivo knock down of miR-153 in the dentate gyrus resulted in an increase in the expression of VAMP2 but not SNAP25

(*Figure 8D,F*). The VAMP2 component of the SNARE pathway may therefore be the major in vivo target for miR-153-mediated regulation of AMPA receptor exocytosis.

The SNARE proteins also play well-established roles in vesicle trafficking, fusion and neurotransmitter release at the presynaptic area (*Rizo and Rosenmund, 2008*; *Südhof, 2013*; *Südhof and Rothman, 2009*; *Weber et al., 1998*). Vamp2 and Snap25 were initially characterized as components of SNARE complexes that mediate vesicle fusion and neurotransmitter release at the presynaptic area (*Söllner et al., 1993a*, *1993b*). In addition, cytoskeleton associated proteins such as Bsn and Pclo (*Fenster et al., 2000*; *Schoch and Gundelfinger, 2006*; *tom Dieck et al., 1998*), both of which are also miR-153 targets (*Figures 1*, *7* and *8*), participate in vesicle transport at the presynaptic area. Our demonstration that miR-153 knock-down increases activity-dependent glutamate release is consistent with a general role for miR-153 in down-regulation of vesicle transport and fusion at both the pre- and post-synaptic areas. In agreement with a role for miR-153 in neurotransmitter release (this study), studies in zebrafish have shown that loss of miR-153 results in increased Snap25 expression and consequently motor neuron defects and hyperactive movement of early zebrafish embryos (*Wei et al., 2013*). At least one other miRNA, miR-137, has been implicated in presynaptic vesicle transport (*Siegert et al., 2015*). MiR-137 overexpression results in the downregulation of presynaptic components of the SNARE complexes, such as Complexin-1 (Cplx1), Nsf, and Synaptotagmin-1 (Syt1), leading to impaired vesicle release. In vivo, overexpression of miR-137 in the dentate gyrus results in changes in synaptic vesicle pool distribution, impaired mossy fiber-LTP induction and deficits in hippocampus-dependent learning and memory (*Siegert et al., 2015*). In contrast to miR-153, miR-137 expression was not induced in our studies. MiR-137 may therefore play an important role in the constitutive regulation of basal SNARE protein levels that impact presynaptic neurotransmitter release.

The molecular machinery required for regulated secretion is conserved across different cell types (*Mostov et al., 2003*). We note that, in addition to the brain, both miR-153 and its host gene, Ptprn2, are expressed in the pancreas and affect insulin secretion following glucose stimulation in pancreatic cell lines (*Mandemakers et al., 2013*; *Xu et al., 2015*). Pancreatic beta cells express many of the components that are required for regulated exocytosis of synaptic vesicles in neurons (*Jacobsson et al., 1994*). In fact, four of the predicted targets from the vesicle exocytosis pathway (*Pclo*, *Snap25*, *Snca* and *Vamp2*) are also regulated by miR-153 in pancreatic cell lines (*Mandemakers et al., 2013*). Therefore, miR-153 downregulates vesicle transport and fusion to suppress neurotransmitter release in neurons and may function in a similar manner to suppress insulin secretion in the pancreas following stimulation with glucose. Moreover, Ptprn2 localizes to synaptic vesicles and facilitates the secretion of neurotransmitters in the brain and insulin in the pancreas through an unknown mechanism (*Nishimura et al., 2010*, *2009*). MiR-153 and its host gene may therefore have the capacity to negatively and positively regulate downstream targets, respectively. However, while in the pancreas miR-153 expression appears to be under the control of the *Ptprn2* promoter (*Mandemakers et al., 2013*), activity-dependent expression of miR-153 in the hippocampus is driven by an alternative promoter (this study), allowing it to act independently of its host gene. Further studies are required to determine the intriguing relationship between the activities of miR-153 and its host gene in neuronal and pancreatic cells.

## Learning-induced miRNAs as barriers to neuronal hyperactivity

In addition to the regulation of the vesicle exocytosis pathway, our bioinformatic analysis revealed that fear-induced miRNAs may negatively regulate other pathways such as the immediate early gene expression program. In this regard, neuronal activity promotes the association of several immediate early transcription factors with the miR-153 promoter, suggesting that miR-153 is likely to be transcriptionally induced by the immediate early gene program in an activity-dependent manner. Similarly to the regulation of vesicle exocytosis pathway, activity-induced miRNAs may function as negative feedback regulators to suppress the expression programs that are required for the initial structural and functional changes at the synapse. MiR-153 and the immediate early gene, CBP/p300, are one example of this type of feedback regulation (*Figure 4F* and *Figure 7—figure supplement 1C*). Our findings suggest that miR-153 may negatively regulate the expression of CBP/p300 as a means of controlling CBP/p300 function and may help to prevent excessive CBP/p300-mediated activation of downstream targets. MiR-153 may act together with other fear-induced miRNAs to provide feedback control to return the immediate early gene program to basal levels of expression.

Experience-dependent rewiring of neural circuits is triggered by changes in activity patterns that initiate and ultimately produce long-term modifications of synapses. We speculate that miR-153, together with other fear-induced miRNAs presented in this study, are induced in response to neuronal activity and may perform at least two functions, which are not mutually exclusive. First, by suppressing many plasticity genes that are induced in activated neurons, late acting miRNAs, such as miR-153, may help stabilize firing rates and restore baseline function, thus enabling the circuit to process new information. Second, miR-153 class of miRNAs may set a threshold for linking the strength of excitatory signals to long-term changes in synaptic strength. For example, ready induction of miR-153 may dampen weak signals but allow changes in synaptic strength as a result of stronger signals. Consistent with the results presented here, other studies have shown that the expression of several mammalian miRNAs is rapidly induced after enhanced neuronal activity coupled to learning in various paradigms, such as contextual fear conditioning or olfaction discrimination (*Gao et al., 2010*; *Smalheiser et al., 2010*). In addition to the dentate gyrus (this study), distinct populations of miRNAs are upregulated in the CA1 region of the hippocampus at early (1–3 hr) and late (24 hr) stages of learning and have been proposed to regulate memory formation by increasing protein synthesis through de-repression of mTOR activity (*Kye et al., 2011*). Thus activity-induced miRNAs are likely to exert both positive and negative effects on memory formation. However, the vast majority of the miRNAs we detected have not been implicated in memory formation and with the exception of miR-9, miR-219 and miR-125a, their role in synaptic plasticity is unknown (*Ifrim et al., 2015*; *Kocerha et al., 2009*; *Malmevik et al., 2016*). Thus, this set of miRNAs is a valuable resource for future studies. Comprehensive functional characterization of miR-153 demonstrated its role as a negative feedback regulator of synaptic strength that is fear- and LTP- induced, primarily in dentate gyrus granule cells. Further studies of fear-induced miRNAs are therefore likely to provide insight into how different regions of the hippocampus respond to learning-induced inputs. Future studies on the molecular function of the fear-induced miRNAs will also shed light on the pathways they target, individually or combinatorially, during memory formation.

## Potential roles for fear-induced miRNAs in neurodegenerative disease

The miRNA pathway has been implicated in the biology of Alzheimer's, Parkinson's, and other neurodegenerative diseases (*Barak et al., 2013*; *Hardy and Selkoe, 2002*; *Lukiw et al., 2012*; *Mouradian, 2012*; *Nelson et al., 2008*). In this regard, miR-153 targets the amyloid precursor protein (*Doxakis, 2010*; *Liang et al., 2012*; *Long et al., 2012*) and alpha synuclein (*Singleton et al., 2003*), which has been proposed to interact with the SNARE machinery and participate in vesicle clustering at the presynaptic area (*Burré et al., 2010*; *Gureviciene et al., 2007*). Moreover, miR-9 dysregulation is frequently associated with Alzheimer's disease and Huntington's disease, miR-338–3p is upregulated in the frontal cortex of human ALS patients (*De Felice et al., 2014*; *Packer et al., 2008*; *Shioya et al., 2010*) and miR29b is downregulated in sporadic cases of Alzheimer's disease and its expression levels are reduced in a murine model of Huntington's disease and human cortex samples from patients with Huntington's disease (*Hébert et al., 2008*; *Johnson et al., 2008*). Altered expression of miR-29b and miR-181b was also found in the postmortem brain of patients with schizophrenia (*Beveridge et al., 2008*; *Perkins et al., 2007*). Activity-dependent induction of these miRNAs may therefore aid in preventing neuronal toxicity by downregulating these and other proteins during neuronal stimulation. In addition to providing mechanistic insight into how fear based memory is encoded and stored in the hippocampus, future functional analysis of the miRNAs presented in this study may prove useful in the development of strategies to treat neurodegenerative diseases.

# Materials and methods

## Materials

The following commercially available antibodies were used for chromatin immunoprecipitation experiments: Anti-H3K4me3 (Millipore, catalog number 04–745; RRID:AB_1163444), Anti-H3K36me3 (Abcam, catalog number ab9050; RRID:AB_306966), Anti-CREB (Millipore, catalog number 06–519; RRID:AB_310153), Anti-PolII (Ser5-phosphorylated) (Abcam, catalog number ab5131; RRID:AB_449369), Anti-PolII (Ser2-phosphorylated) (Abcam, catalog number ab5095; RRID:AB_304749), Anti-CBP/p300 (Millipore, catalog number 05–257; RRID:AB_11213111), Anti-C/EBPß (Santa Cruz,

catalog number sc-150; AB_2260363) or Anti-ATF4 (Santa Cruz, catalog number sc-200; RRID:AB_2058752). The following commercially available antibodies were used for microscopy experiments: anti-RFP (Invitrogen R10367; RRID:AB_2315269), anti-GFP (Aves Labs GFP1010; RRID:AB_2307313), goat anti-rabbit Alexa 555 (Invitrogen A21428; RRID: AB_10561552) and goat anti-rabbit Alexa 488 (Invitrogen A11039; RRID:AB_142924). The following cell lines were used: rat H19-7 (ATCC CRL-2526; RRID: CVCL_H781), mouse Neuro-2a (ATCC CCL-131; RRID:CVCL_0470), human HEK-293FT (Invitrogen R70007; RRID:CVCL_6911), human HEK-293T (ATCC CRL-3216; RRID:CVCL_0063).

## Methods

All experiments were performed according to the Guide for the Care and Use of Laboratory Animals and were approved by the National Institutes of Health, the Committee on Animal Care at Harvard Medical School (Boston, MA, USA), the Committee on Animal Care at the Massachusetts Institute of Technology (Cambridge, MA, USA) and the Committee on Animal Care at the University of Vermont (Burlington, VT, USA).

### Fear conditioning (rats)

Contextual fear-conditioning with naïve controls

The subjects were 24 female wild-type Wistar (RRID:RGD_2308816) rats. Sex differences in laboratory rodent species have been reported during acquisition, retention and extinction in classical and operant conditioning paradigms (*Maren et al., 1994*). In the classical fear-conditioning paradigm males outperform females, or are more resistant to extinction (*Dalla et al., 2009*). Typically behavioral training does not include cohorts of female rodents because it is assumed that their behaviors are similar, but more variable, relative to males (*Dalla et al., 2009*). However, we confirmed that overexpression of miR-153 after fear conditioning is not a sex-specific response by performing behavioral training with adult male wild-type mice. We observed an increase in hippocampal levels of miR-153 following training (data not shown). Rats were approximately 90 days old at the start of the experiment and were individually housed with ad lib food and water throughout the experiment. Two sets of four operant chambers (Med-Associates, St. Albans, VT) located in separate rooms were used (counterbalanced). Chambers from both sets measured $31.75 \times 24.13 \times 29.21$ cm ($l \times w \times h$) and were individually housed in windowed sound attenuation chambers. Ventilation fans provided background noise of 65 dB, and the boxes were lit with two 7.5 W incandescent bulbs mounted to the ceiling of the sound-attenuation chamber. In one set of chambers, the front and back walls were brushed aluminum; the side walls and ceiling were clear acrylic plastic. Recessed $5.1 \times 5.1$ cm food cups were centered in the front wall and positioned near floor-level. The floor was composed of stainless steel rods (0.48 cm in diameter) spaced 1.6 cm apart from center to center and mounted parallel to the front wall. In the second set of chambers, the ceiling and left side wall had black horizontal stripes, 3.8 cm wide and spaced 3.8 cm apart. The floor consisted of alternating stainless steel rods with different diameters (0.48 and 1.27 cm), spaced 1.6 cm apart from center to center. The ceiling and left sidewall were covered with rows of dark dots (1.9 cm in diameter) that were separated by approximately 1.2 cm. The US was a 2 s, 1.2 mA shock provided by Med Associates shock sources.

The rats were handled daily for 5 days prior to the start of the experiment. On Day 6, fear was conditioned to the context for 12 rats. Rats were placed in the conditioning chamber for one 5 min session. Two minutes after being put in the box, the first of 3 shock (1.2 mA, 2 s) USs was delivered. The second shock was delivered 60 s after the first, and the third shock was delivered 60 s after that. The naïve group was moved into the laboratory and back to the colony room to control for handling. On Day 7 (exactly 24 hr after exposure to the fear conditioning paradigm) 4 experimental rats (Trained) and 4 naïve rats (Naïve) were placed back in the boxes for 5 min and tested for context freezing. This control group of 8 rats was used to assess behavior after exposure to the contextual fear conditioning paradigm; tissue was not harvested from these animals (*Figure 1A*). Exactly 24 hr after exposure to the fear conditioning paradigm, 8 experimental rats (Trained) and 8 control rats (Naïve) were euthanized and the hippocampus was isolated for subsequent analysis. Freezing was scored from videotape with a time-sampling procedure in which the rat's behavior was scored as freezing or not every 3 s. Freezing was defined as the absence of all movement, except for that related to breathing (*Fanselow, 1980*). The percentage of all samples spent freezing was calculated

for each rat. A second observer (blind to treatment) scored all data from testing. The correlation between observers scores was $r(14) = 0.98$.

## Contextual fear-conditioning with immediate shock controls

The subjects were 27 female wild-type Wistar (RRID:RGD_2308816) rats. Rats were approximately 90 days old at the start of the experiment and were individually housed with ad lib food and water throughout the experiment. Two sets of four operant chambers (Med-Associates, St. Albans, VT) located in separate rooms were used (counterbalanced), as previously described in our methods section. The rats were handled daily for 5 days prior to the start of the experiment. On Day 6, fear was conditioned to the context for 9 rats (Trained). Rats were placed in the conditioning chamber for one 5 min session. Two minutes after being put in the box, the first of 3 shock (1.2 mA, 2 s) USs was delivered. The second shock was delivered 60 s after the first, and the third shock was delivered 60 s after that. A second group of 9 rats was trained with an immediate shock procedure. Rats were placed in the conditioning chamber for one 5 min session. Each rat was placed in the conditioning context and 5 s later the first of 3 shock (1.2 mA, 2 s) USs was delivered. The second shock was delivered 2 s after the first, and the third shock was delivered 2 s after that. The naïve group was moved into the laboratory and back to the colony room to control for handling. On Day 7 (exactly 24 hr after exposure to the fear conditioning paradigm) 4 delayed shock (Trained) rats, 4 immediate shock rats and 4 control (Naïve) rats were placed back in the boxes for 5 min and tested for context freezing. This control group of 12 rats was used to assess behavior after exposure to the contextual fear conditioning paradigm; tissue was not harvested from these animals. Exactly 24 hr after exposure to the fear conditioning paradigm, 9 delayed shock rats (Trained), 9 immediate shock rats and 9 control rats (Naïve) were euthanized and the hippocampus was isolated for subsequent analysis. Freezing was scored from videotape with a time-sampling procedure in which the rat's behavior was scored as freezing or not every 3 s. Freezing was defined as the absence of all movement, except for that related to breathing (*Fanselow, 1980*). The percentage of all samples spent freezing was calculated for each rat. A second observer (blind to treatment) scored all data from testing. The correlation between observers scores was $r(12) = 0.99$.

## miRNA expression analysis

For measurements of miR-153 expression: From cultured neurons, and from brain tissue, small RNA was isolated from homogenized tissue using the mirVana miRNA Isolation Kit (Ambion). From hippocampal tissue from trained and un-trained animals, total RNA was isolated using the RecoverAll Total Nucleic Acid Isolation Kit (Ambion). 1 µg of total RNA was used for analysis using the TaqMan Small RNA Assay (Taqman) designed specifically for miR153 (mmu-miR-153) and the control (snoRNA202). All analysis and controls were run in triplicates and data is presented as mean ± SEM. For brain tissue, three different tissues from three different animals were analyzed. For hippocampal tissues, three different tissues from three different trained or untrained rats were analyzed in triplicate. Data are presented as mean ± SEM.

For measurements of small RNA expression for the immediate versus delayed shock animals, hippocampal tissue from immediate shock, delayed shock (trained), and naïve rats (n = 9 per condition) was homogenized and small RNA was isolated using the miRVana miRNA isolation kit (Ambion). Small RNA from the nine animals was pooled together into three groups to generate three biological replicates for each condition (immediate shock, delayed shock, and naïve). Three 500 ng pools of small RNA per condition were used for RT-qPCR analysis with the TaqMan small RNA assay (Applied Biosystems) with TaqMan RT and PCR primers designed specifically for each miRNA as well as the control (RNU58). Each of the three pools of small RNA from nine different trained, immediate shock or naïve rats were analyzed in triplicate. Expression values were first normalized to the control (RNU58) then divided by naïve values to determine fold-change relative to the naïve condition. Data are presented as fold-change relative to naïve rats from the three pools measured in triplicate (mean ± SEM).

## Acute hippocampal slice preparation and electrophysiology

P25-30 wild-type mice were decapitated and the hippocampal lobules were rapidly isolated in artificial cerebral spinal fluid (aCSF). aCSF contained (in mM): 119 NaCl, 2.5 KCl, 1 $NaH_2PO_4$, 26.3 $NaHCO_3$, 11 glucose, 1.3 $MgSO_4$, and 2.5 $CaCl_2$. Transverse slices (400 µm) of the hippocampus were then cut

using a tissue chopper (Stoelting). Slices were then incubated in oxygenated aCSF at room temperature for at least 1 hr before recording. Then, slices were transferred to a recording chamber, maintained at 32°C and continuously perfused at 1–2 ml/min with oxygenated aCSF. Recording electrodes were pulled from borosilicate capillary glass and filled with 3 M NaCl (1.5 mm o.d.; Sutter Instruments). The recording pipette was placed in the medial molecular layer of the dentate gyrus. Recordings were made with a MultiClamp 700 B amplifier, collected using Clampex 10.3, and analyzed using Clampfit 10.3 (Molecular Devices). Field excitatory postsynaptic potentials (fEPSPs) were evoked using cluster electrodes (FHC) also placed in a medial molecular layer of the dentate gyrus. Current between 0.1 and 1 mA for 0.1 ms was delivered every 30 s with a stimulus isolator (World Precision Instruments). For experiments, current was set at a level to elicit 30% of the maximum response. After 20 min of stable baseline, LTP was induces by delivering 100 pulses at 50 Hz (four times separated by 15 s). 3 hr after LTP induction slices were harvested for further analysis. Control slices were from the same hippocampus and were incubated for the same period of time in oxygenated aCSF.

## Stereotactic injections

Stereotactic injections into the dentate gyrus of adult male mice were performed as described (*Cetin et al., 2006*) with the following modifications: eight week old C57BL/6 mice were bilaterally injected with 1 μL lentivirus mixed with 0.2 μL Fast Green dye and overlaid with mineral oil in a glass micropipette (Drummond Wiretrol 10 μL). The stereotactic coordinates were anterior/posterior: −2, medial/lateral: 1.6, dorsal/ventral: −1.65. The total injection volume was 1 μL, injected at a rate of 0.125 μL/min. 3–4-months-old miR153KD and control mice were used for experiments. For consistency purposes, only male mice were used in all experiments.

## Mouse behavior

All the behavior experiments were performed using groups of 8–11 3–4-month-old male miR153KD (mutant) and scramble KD (control) injected mice. Mice were housed in groups of 3–5 animals. Mice were left to acclimate in the testing rooms for 45 min prior to the experiments. If the same groups of mice were used in different behavioral experiments, the tests were separated by one week. The most stressful test, fear conditioning, was performed last. All the experiments were done during the light phase, the second part of the day, with experimenters that were blind to the genotype and treatment of the mice.

### Open field

Activity in a novel environment was measured in the 40 × 40 cm Plexiglas VersaMax chambers (Accuscan Instruments, Columbus, OH) using sets of 16 photobeam arrays. During each 60 or 10 min session, the number of beam breaks (activity) was measured automatically. The animal's position was automatically determined and the mouse was tracked for 10 min with VersaMax software (TSE systems, Chesterfield, MO).

### Light/dark exploration

The light-dark apparatus consisted of a transparent Plexiglas open field box (40 × 40 cm) containing a black Plexiglas box (20 × 20 cm) occupying half of the area. Experiments were conducted in a room with the overhead light off, and a bright 120 W lamp directed at the light part of the open field area. A mouse was placed in the black Plexiglas box and its behavior was video recorded for 10 min. The frequency of exits to the bright area, and the time spent in the bright and the dark areas were scored.

### Hot plate analgesia test

Animals were placed on a plate set to 55°C and the timer was started. The animals were observed until they start showing nociceptive response (rear paw licking) and the latency to respond was recorded. Hot plate analgesia test meter (IITC Life Sciences Inc., CA) was used for the test.

### Contextual and cued fear conditioning

For fear conditioning experiments, the TSE fear conditioning system (TSE systems) was used. During contextual fear conditioning tests, mice were placed in a conditioning chamber with Plexiglas walls and a metal grid bottom. They were left to acclimate for 3 min and were then foot-shocked (2 s, 0.8

mA constant current). After 24 hr in the home cage, mice were returned to the same chambers and the freezing bouts, defined as a total lack of movement except for a heartbeat and respiration, were scored during every 10 s during a 3 min period. Cued fear conditioning was performed by placing the animals in the test chamber for 3 min following the exposure to the auditory cue (30 s, 20 kH, 75 db sound pressure level) and a foot shock (2 s, 0.8 mA, constant current). Associative learning was assessed 24 hr later by placing the mice into the modified chambers (visual, tactile, and olfactory changes) and delivering the identical auditory cue for 3 min. Freezing behavior was recorded as described above.

## Cell cultures and transfection

Cultures of dissociated primary hippocampal neurons from embryonic day 17 (E17) Swiss Webster wild-type mice (Charles River laboratories) were prepared as described (Lin et al., 2008). They were maintained in Neurobasal medium supplemented with B27 and N2 supplements (Invitrogen), penicillin-streptomycin (50 µg/ml penicillin and 50 U/ml streptomycin, Invitrogen) and Glutamine (1 mM, Invitrogen). For biochemical experiments neurons were plated at high density (125.000–150.000 cells/cm$^2$) and for imaging at lower density (up to 100.000 cells/cm$^2$) on poly-L-lysine coated multi-well dishes. At DIV3 (5 µM) final concentration of cytosine-b-D-arabinofuranoside was added into the cultures to inhibit glial cell proliferation and the cultures were fed every 3 days from there on. Neuronal transfections were performed with lipofectamine 2000 reagent (Invitrogen) by incubating the plasmids with neuronal cells for a short time to reduce cell death.

HEK293 cells were maintained in DMEM medium (Invitrogen) plus 10% FBS (Invitrogen), 1 mM Glutamine and 100 µg/ml penicillin-streptomycin following standard culture conditions.

## Stimulation of neuronal cells

KCl-mediated depolarization of neurons was achieved following previously established methods (Tao et al., 1998). Briefly, to induce neuronal activity hippocampal cultures were treated for 2 hr with 1 µM tetrodotoxin (TTX) (Tocris), and 100 µM (2R)-amino-5-phosphonopentanoate (DL-APV) (Tocris) to reduce spontaneous neuronal activity. Then, neurons were depolarized by the addition of 31% depolarization buffer (170 mM KCl, 2 mM CaCl2, 1 mM MgCl2, 10 mM HEPES) to the media for the indicated lengths of time.

## Gene expression analysis

For measurements of gene expression in cultured neurons, total RNA was isolated using the RNeasy mini kit (Qiagen). 1 µg of total RNA was digested with DNAse I (Invitrogen, Cambridge, USA), and cDNA was produced using oligo dT primers and the Superscript III First-Strand Synthesis Kit (Invitrogen). The amount of each transcript was measured using gene specific primers obtained from (Applied Biosystems) and the Taqman gene expression system (Applied Biosystems). Gene-specific measurements and standards were performed in triplicate.

## Ontological and canonical signaling pathway analysis

A complete list of downstream targets for all 21 miRNAs identified in *Figure 1B* was compiled from the miRBase (RRID:SCR_003152), microrna (RRID:SCR_006997) and TargetScan (RRID:SCR_010845) databases. The list was reduced to a total of 3986 downstream targets after excluding targets that are not expressed in the brain using the Partek Genomics Suite (http://www.partek.com/pgs; RRID:SCR_011860). From this list, a total of 353 downstream targets were identified that possess three or more seed sequence matches for any combination of the 21 miRNAs identified in this study (*Supplementary file 1A*).

Ontological analysis used Gene Ontology (GO) categories to determine processes or functional categories that were represented in the combined downstream target list, using the canonical pathway module of MetaCore (http://thomsonreuters.com from Thomson Reuters, New York; RRID:SCR_008125). This analysis determined the number of genes for a given network that are present in the downstream target gene list and the number of genes that would be part of that category by random chance given the number of commonly expressed genes. Statistical significance of each canonical signaling pathway was established by p-value cutoff (p-value<0.01). Only the processes or categories which passed this threshold were used to determine the list of genes used to identify the

networks presented in *Figure 1C*. The statistical significance of each canonical signaling pathway presented in *Figure 1C* was further established by comparison to canonical signaling pathways identified from random sets of brain-expressed genes. A list of 20,000 brain-expressed genes was used to generate random sets of genes for network analysis. Random sets of genes were created that were similar in size to the target gene list (3986) and network analysis was performed with each random gene list. The statistical significance reported in *Figure 1C* reflects the statistical significance for each canonical signaling pathway as compared to the number of occurrences from random sets of brain-expressed genes.

## Virus production and infection

Viruses targeted against miR-153 (Lv-miR-153 KD), a scrambled control (Lv-scramble KD) were created by constructing a lentiviral expression vector using the miRZIP shRNA lentivector from a parent vector pGreenPur (System Biosciences). A miR-153 overexpressing vector (Lv-miR-153 OE) and a scramble overexpressing vector (Lv-scramble OE) based on the pCDH lentivectors (SBI) with a cassette for GFP as reporter. Plasmids were purified using a mega-prep DNA isolation Kit (Qiagen). For viruses packaging we used the Virapower system (Invitrogen) and it was performed in 293FT cells. Medium containing the viral particles was collected 72 hr after transfection and viral particles were concentrated using the PEG-it Virus precipitation solution. Viral titer was determined using SBI's Ultra Rapid Lentiviral Titer Kit. Viral transduction of neurons was achieved using a multiplicity of infection (MOI) of at least 20 for all different conditions in combination with Transdux at DIV11 for both the knockdown and overexpression experiments. Virus production, packaging, purification, and transduction were achieved following protocols (System Biosciences). Viral expression was analyzed using QuantiMir RT Kit Small RNA Quantitation System (System Biosciences) using primers specific to the virus.

## Luciferase assay

MiR-153 target sequences were analyzed using miRBase, microrna and TargetScan (*Betel et al., 2008*; *Griffiths-Jones et al., 2008*; *Lewis et al., 2005*). Gene specific 3'-UTR constructs containing known miR-153 target sequences were amplified by proofreading RT-PCR from adult mouse hippocampus total RNA by using the primers specific to each 3'-UTR element (primer sequence information is presented below). The PCR product was cloned into the pMIR-REPORT vector (Ambion). An expression vector directing the synthesis of mmu-miR-153 (miRBase accession no. MI0000175) and a scramble control sequence were prepared as previously described (*Doxakis, 2010*). All vectors were checked by sequencing before use. HEK293T cells were transfected in a 1:1:1 ratio with a single 3'-UTR luciferase construct, a control pRenilla vector, and either the miR-153-eGFP expression construct or the Scrambled-eGFP expression construct using lipofectamine 2000 (Invitrogen). Targeted knockdown was analyzed using the Dual-Luciferase Reporter Assay System (Promega). Knockdown activity by miR-153-eGFP was controlled to the Scrambled-eGFP. Data are presented as mean ± standard deviation.

| Gene | Forward primer | Reverse primer |
|------|---------------|----------------|
| *Bsn* | CTTTCAAGAGACCCTGCCTTAC | CCCTATGAAGTGAGTGTGTTGAG |
| *Pclo* | CATGACTGTGGGATACAAAGAGA | CAGTATTTATTAGTAAGGCTGGTACAAC |
| *Snap25* | CTGTGCTCTCCTCCAAATGT | TCGGTGGCTGTGATCTATAATTT |
| *Snca* | AAGAATGTCATTGCACCCAATCTCC | AATATTATCCATTGCAAAATC |
| *Trak2* | CCACTAACTGACCTCGTGTATAA | AAGCAAAGGAAGGTGCATAAAG |
| *Vamp2* | AGTCTGCCCTGCCTAAGA | CTGGATGCGCCACAGAAT |

## Primer sequences for 3'-UTR cloning into luciferase reporter vector pMIR-REPORT

3'-UTR luciferase constructs with positions 4–6 of the seed sequence mutated to ATT, were prepared for each gene (*Bsn, Pclo, Snap25, Snca, Trak2,* and *Vamp2*) by site-directed mutatgenesis. All mutated vectors were checked by sequencing before use. Luciferase assays were performed as

described above. Knockdown activity by miR-153-eGFP was reported relative to scrambled-eGFP. Data are presented as mean ± standard deviation.

## Microarray

Total RNA was isolated from the hippocampus of trained and naïve rats using the miRVana Kit (Ambion). A total of 5 μg for each sample was shipped to Miltenyi Biotec on dry ice. Quality control of total RNA and a pool of total RNA from three separate samples for each condition (naïve or trained) was labeled and hybridized to the miRXplore microarray. Quality control, sample labeling, hybridization and data analysis were performed by Miltenyi (www.miltenyibiotec.com). Samples A and B presented in *Figure 1—figure supplement 1* represent rats from a single contextual fear conditioning experiment; samples C and 1–4 represent rats from an independent contextual fear conditioning experiment (see Fear Conditioning for rats in Supplemental Materials and methods for a full description of experimental procedures).

Pearson and Spearman correlation coefficients were calculated for the top 21 miRNAs identified from the three biological replicates of the miRNA microarray. For comparison, Pearson and Spearman correlation coefficients were calculated for a random group of 21 miRNAs selected from the miRNAs included in the miRXplore microarray. These values are presented in the tables below:

## Pearson coefficient

| Microarray replicates | 21 fear-induced miRNAs | 21 random miRNAs |
|---|---|---|
| Group A and B | 0.31 | −0.17 |
| Group A and C | 0.12 | 0.03 |
| Group B and C | 0.62 | 0.24 |

## Spearman coefficient

| Microarray replicates | 21 fear-induced miRNAs | 21 random miRNAs |
|---|---|---|
| Group A and B | 0.31 | −0.10 |
| Group A and C | 0.12 | 0.04 |
| Group B and C | 0.62 | 0.11 |

## Microscopy in neurons

For phenotypic analysis hippocampal neurons were grown on No. 1.5 12 mm glass coverslips (Electron Microscopy Sciences) coated with 0.1 mg/ml poly-L-lysine overnight at 37°C and plated at a density of 90,000–100,000cells/cm$^2$ in 24-well dishes. The cells were cultured as mentioned before and at DIV11 they were transfected with the vectors indicated in the figures and the text. Cultures were fixed at DIV18 with 4% formaldehyde/2% sucrose in 1x phosphate-buffered saline (PBS), washed 3 times with 1x PBS and then mounted with Prolong Gold anti-fade reagent (Life Sciences). Lifeact-Ruby neurons were identified by anti-RFP primary antibody and then a secondary Alexa-555. This procedure was similar to that used for dendritic spine geometry analysis. Neurons for the phenotypic analyses were imaged using either widefield or spinning disk confocal microscopy. Widefield images used for the phenotypic analyses were collected using a Nikon Ti-E inverted microscope with a Hamamatsu ORCA R2 cooled CCD camera controlled with MetaMorph 7 software (RRID:SCR_002368). GFP was imaged with a 480/40 excitation filter and 535/50 emission filter, Lifeact-Ruby was imaged with a 545/30 excitation filter and a 620/60 emission filter and DAPI was imaged with a 350/50 excitation filter and 460/50 emission filter. Spinning disk confocal images used for figures were collected using a Yokogawa CSU-X1 mounted on a Nikon Ti-E, a Spectral Applied Research LMM-7 laser launch with AOTF control of intensity and wavelength, and a Hamamatsu ORCA-AG cooled CCD camera controlled with MetaMorph 7 software. GFP was imaged with a 491 nm solid state laser

and a 535/50 emission filter, and Lifeact-Ruby neurons were imaged with a 561 nm solid state laser and 620/60 emission filter, both using a QUAD 405/491/561/642 dichroic. All filters were made by Chroma Technologies. Images for dendritic spine analysis were acquired using a Nikon Plan Apo 60 × 1.4 NA oil immersion objective. For each image obtained, 15 focal plane z-series were collected with a step size of 0.25 μm using the internal Nikon Ti focus motor. Images for dendritic analysis were acquired using Nikon Plan Apo 20 × 0.75 NA objective lens. For each hippocampal neuron imaged, 10 focal plane z-series were collected with a step size of 1 μm. Images in the figures are displayed as maximum intensity z-projections using MetaMorph 7 or ImageJ (RRID:SCR_003070).

## Dendritic spine analysis

To analyze the effects of miR-153 on dendritic spine shape, we analyzed in a blinded manner totally 14–16 neurons for miR-153 inhibition and 19–23 neurons for miR-153 overexpression. Neurons were selected based on the eGFP staining. For this analysis, we obtained images from two independent experiments per condition and three coverslips for each experiment. Filament tracer plugin of Imaris software (version 7.6.5, Bitplane Inc.; RRID:SCR_007370) used for this analysis. Similar to the dendritic analysis, in the Imaris Surpass mode, a new 3D filament was created using the Autopath mode and a region of interest (ROI) was selected. We restricted the spine analysis in secondary or tertiary dendrites to reduce variability and we analyzed about 100 μm–150 μm of dendritic segments from at least two dendrites per neuron. The dendrite reconstruction was created based on the eGFP signal and dendritic spines were reconstructed based on the Lifeact-ruby signal. After the completion of the analysis for each neuron we extracted the data for the spine volume, spine length, spine terminal diameter (head width), spine mean neck width. Spine density was calculated per neuron by dividing the total number of spines by the total length of dendrites measured in each cell. The means for these parameters were calculated for each neuron separately and then the averages from all neurons together were plotted as average numbers for each condition. Alternatively, all the individual values were used to generate cumulative plots. In addition to the unpaired t-tests, we performed multiple pairwise comparisons using one-way ANOVA: [spine volume: (p value = 0.04, F = 2925), spine width: (p value = 0.0124, F = 3904), spine length: (p value = 0,4231, F = 0,9476), spine neck width: (p value = 0,0046, F = 4741), spine density: (p value = 0,0154, F = 3731)].

## Microscopy of brain sections

To assess the efficiency of the virus injections mice were transcardially perfused with phosphate-buffered saline (PBS) followed by 4% paraformaldehyde (PFA) in 1x PBS. The brains were dissected and post-fixed in 4% PFA in PBS at 4°C overnight. Free-floating vibratome coronal sections (40 μm) were incubated in a blocking solution of 10% normal donkey serum, 3% bovine serum albumin, 0.2% Triton-X 100, 0.02% sodium azide in 1x PBS for 1–2 hr at room temperature (RT). Sections were then incubated with primary antibodies in the blocking solution overnight at 4°C followed by the appropriate Cy3-congugated (Jackson Labs, ME; 1:1000) and Alexa488 (Invitrogen, OR; 1:1000) secondary antibodies for 2 hr at RT. Twenty minute incubations with Hoechst dye (Invitrogen, OR) at RT were performed to label cell nuclei. GFP-expressing neurons were identified with anti-GFP chicken polyclonal antibody (GFP-1010, Aves Labs) at 1:1000. Images were acquired using high-resolution multichannel scanning confocal microscopy (LSM 510 Imager Z.1; Zeiss). Confocal 3D scans were carried out with a using Plan-Apochromat 63×/1.4, EC Plan-Neofluoar 40×/1.30 oil immersion, and Plan-Apochromat 20×/0.8 objective lenses at four excitation laser lines. DAPI was imaged with a 405 nm solid state laser and a 445/50 emission filter.

## Chromatin immunoprecipitation

The Chromatin Immunoprecipitation assays were performed as described by (Tatarakis et al., 2008). Briefly, to crosslink chromatin the cells were treated with 1% formaldehyde for 10 min at room temperature. Crosslinking was stopped by the addition of glycine to a final concentration of 125 mM. The cells were washed with 1x PBS and nuclei were prepared by resuspension in a sucrose buffer [0.32 M sucrose, 15 mM Hepes pH 7.9, 60 mM KCl, 2 mM EDTA, 0.5 Mm EGTA, 0.5% BSA, 0.5 mM spermidine, 0.15 mM spermine, 0.5% NP-40 and 0.5 mM DTT] followed by dounce homogenization. The nuclei were lysed in sonication buffer [45 Mm Hepes pH 7.9, 110 mM NaCl, 5 Mm EDTA, 1% Triton X-100, 0.3% SDS, 0.1% Na-deoxycholate, and protease inhibitor cocktail (Roche)]

and then sonicated with the Bioruptor for 20 min (30 s on 30 s off). After centrifugation the soluble chromatin was precleared with dynabeads and subjected to IPs. Reverse crosslinking and DNA purification was following the immunoprecipitations. After centrifugation the soluble chromatin was precleared with dynabeads and subjected to immunoprecipitation with one of the following antibodies: H3K4me3, H3K36me3, CREB, PolII (Ser5-phosphorylated), PolII (Ser2-phosphorylated), CBP/p300, C/EBP ß or ATF4. Reverse crosslinking and DNA purification followed the immunoprecipitations. The immunoprecipitated DNA was analyzed by qPCR. Tiling primers were designed against the mouse genome for each H3K4me3 peak and across the miR-153 coding sequence; with each primer pair spanning a 400 base pair region (sequences are presented below).

## Ptprn2 promoter tiling primer set (chr12: 117,723,700–117,725,700)

| Primer ID | Forward sequence | Reverse sequence |
|---|---|---|
| R1 | TTTGAGGACTCCATCTGCAACTCC | GTTCGTGGAGAAAGGACACTTGGA |
| R2 | TGCTGCTGCTGCTGCTAC | TTGAGCTGTCCCAGGTCCTT |
| R3 | TTAGTGAGTGGCTGGGTCCTT | ACAGACAAGATTAGCAGGAGGGAG |
| R4 | CCTCTGGTCTTCAGAGGTGTTTCA | TTCAAGGAGTCTCATGTGGTAGGC |

## Ptprn2 cryptic promoter tiling primer set (chr12: 118,392,000–118,395,000)

| Primer ID | Forward sequence | Reverse sequence |
|---|---|---|
| R1 | GAGGCTTCATTCCCTCACCCTAAT | CCTCATCCGCCCAAGACTATGAAT |
| R2 | TCGCAGAACTGCCTGCAC | CAACCAACTGCTTCCCTGCATT |
| R3 | TTCAATGCAGGGAAGCAGTTGG | CAACCCTTGGAAGGTTCTGTTCTG |
| R4 | CGTCTTGCCTAGTTCAGAGGGTAA | TGTCTCTCCCTTCTAATCTGTGCG |
| R5 | GGGAGGATGGATGAAGGACAAGAT | GGGAGGTAGAAGCTCAAAGGTGAT |
| R6 | GATCCTACACTTTCTCCCACCCTT | CAACGAAGACGCAAAGGGACTT |
| R7 | GTGGTAGAACTAGGTGTGTACTGC | CGCAATGCCTGGTACCTAAGAAAG |
| R8 | CCGATCTGGTGTAAAGGGCTTAGT | GTGCACTTTAGGAGTGGAGCATCT |
| R9 | ATGGTCTGATCTCCACGACCTCTA | CTTAACTCGCTCTCATGCCCGTAA |
| R10 | GTACGGCCAAATATCCCTCTCCAA | ATCTATCCAAAGAGGGAACCTGCC |

## Ptprn2 control region tiling primer set (chr12: 118,382,000–118,385,000)

| Primer ID | Forward sequence | Reverse sequence |
|---|---|---|
| R1 | AGACAGCTACATCTGGGTCCTTTC | ACACCCATGGAAGGAGTTACAGAG |
| R2 | TCTCTGCCATTCGGTATTCCTCAG | GCAATGGCTTGTGCTGTAAGATCG |
| R3 | GGGCCTTGAATTCCTGATCTTTCC | CTCGCTAACCGAATCCAAGAACAC |

## Ptprn2 miR-153 coding sequence tiling primer set (chr12: 118,487,290–118,491,358)

| Primer ID | Forward sequence | Reverse sequence |
|---|---|---|

| | | |
|---|---|---|
| R1 | GTATGCTCACTTGTGTCCTTCTGC | CATGACCCACACTTCTGACTTCAC |
| R2 | TGTCTGGATGATCAGTGTAAGGTGAC | CCAAGCCTTTGTAAATCAACCCGC |
| R3 | CAACTCAAGCAAATAGCAGCCTCC | GACGCTAAATTACAGGCAGCAGTG |
| R4 | CGTGGTTCTCATCCCAGGGAAATA | AGGCAATGTGTGTGTGCTGAATC |
| R5 | ATGGTCATGATAACACCCAGGCTC | TGAGTGTAGCTAACTGAGCTGTGC |
| R6 | TCGCTCATGAGTCAACTCCTCTTC | GGTGGAAGGTCTCTGTGAGTGAAT |
| R7 | CTTCAGCCTCTCCCATACTGAACA | ATGGGAAGTGAAGACTGGAGACAG |
| R8 | CAGATGACCTTGGACACACAGAGA | TCGCTAGTCACAAACTGGACCTAC |

Chromatin immunoprecipitation assays were also performed with naïve and trained rat hippocampal tissues as described above with the H3K36me3 antibody. Tiling primers were designed against the rat genome across the miR-153 coding sequence; with each primer pair spanning a 400 base pair region (sequences are presented below).

## Ptprn2 control region tiling primer set (chr6: 137,500,000–137,503,000)

| Primer ID | Forward sequence | Reverse sequence |
|---|---|---|
| R1 | CAGTCCTGGCAATGCTTCTA | CTGTGTGGATCACTGTCTCTTC |
| R2 | GCCACAGAGGAATGCTACTT | AAGTTGGTGCCGGTGTATAG |
| R3 | TTAAAGGGCCACGGTGTTAG | GTGTAGAGGGACCAAGAAGAAG |

## Ptprn2 miR-153 coding sequence tiling primer set (chr6: 144,519,037–144,522,600)

| Primer ID | Forward sequence | Reverse sequence |
|---|---|---|
| R1 | GGCTGGGATGGTTGGTTAAT | CTGCTCTTGACTCTTCCAGATG |
| R2 | CTGGTCAGGGATAGGGAGAATA | CTCTCCCTCCATTGACATACAC |
| R3 | TCAAGTGGCTCAGGATCTTTG | GCTTTGGCCATAGTGTTTCATC |
| R4 | GCTCTGCCTGCTTCCTTATAG | GTGGAGGTCAGAAGTCAATGTAG |
| R5 | CCCACACACACCACACATTA | GATCAGTGCGTGAGCCAATA |
| R6 | GGTGTGATGAAGACAGACAAGA | AAGCAGTGAACTCTCCCATTAG |
| R7 | GTGGAGTTCATGGAGGGAATAG | GTGAGACAGGTCAGAAGGAAAG |
| R8 | ATCAGAAGACGGAGGTGTAATG | CCACACCCTCAATACTGTAACT |
| R9 | GACCATTCCTTCACTGGCATTA | TCCAGACTGTCGAAGTTCTCT |

## Single-cell mRNA expression profiling

Stereotactic injections of miRZip-153 and miRZip-scr viruses into the dentate gyrus of adult male mice were performed as described above. Hippocampal and cortical tissue was separated from remaining brain tissue for both miRZip-153 and miRZip-scr animals and manually disrupted using a sterile razor blade down to ~1 mm$^3$ pieces. The tissue was then dissociated into a single-cell suspension using the trypsin Neural Dissociation Kit (Miltenyi Biotec) according to manufacturer's instructions. Cells were placed into FACS pre-sort medium (Neurobasal medium, 0.25% HEPES, 0.5% FBS).

GFP$^+$ and GFP$^-$ cells were sorted by FACS into skirted 96-well PCR plates containing Pre-Amplification solution (Cells Direct kit, Life Technologies) and appropriate mixtures of TaqMan assays (for mouse). Plates were transported on ice and briefly centrifuged before pre-amplification (94°C 10 min, 50°C 60 min, 94°C 30 s, 50°C 3 min × 15 cycles). Target-specific cDNA from 100 cells per condition (miRZip-153 or miRZip-scr) were harvested, screened for expression of housekeeping genes

*ACTB* and *GAPDH* and then used for expression profiling with a panel of qRT-PCR assays specific to miR-153 and the vesicle exocytosis downstream target genes (*Bsn*, *Pclo*, *Snap25*, *Snca*, *Trak2*, and *Vamp2*). A total of 900,000 cells were obtained from each condition (miRZip-153 or miRZip-scr) and 10% of this population was GFP$^+$ for each condition.

## Fluorescence recovery after photobleaching (FRAP)

Neuroblastoma N2A cells were grown in DMEM media with 25 mM Hepes (pH 7.2) and without phenol red on multi-well glass bottom plate with high performance #1.5 cover glass. They were co-transfected with the plasmid pCI-SEP-GluA1 to express the GluA1 subunit of AMPA receptors tagged with Super Ecliptic pHluorin (SEP), along with pCDH lentivectors (SBI) in which GFP was replaced by mCherry, to overexpress miR-153 or a scramble control that expresses the mCherry to visualize transfected cells. One day before imaging they were differentiated with serum deprivation. Wide field images were acquired using a Nikon Ti-E motorized inverted microscope equipped with Plan Apo 100x NA/1.49 objective lens and the Perfect Focus System for maintenance of focus over time. Images were collected with a Hamamatsu D2 cooled CCD camera controlled with MetaMorph 7 software. SEP-GluA1 was imaged with a 480/40 excitation filter and 535/50 emission filter, mCherry was imaged with a 545/30 excitation filter and a 620/60 emission filter. Photobleaching of cells was performed using the 488 laser TIRF illumination light path in epifluorescence mode with 100% laser intensity for 45 s followed by widefield imaging for the remainder of the experiment. The integrated SEP intensity of photobleached cells (N = 5–8 cells/condition) at each time point was measured using Fiji software, and normalized to that of un-bleached cells to correct for possible fluorescence decay due to repetitive image acquisition. The proportion of recovered SEP at each time point after bleaching (corrected fluorescence intensity/pre-bleaching fluorescence intensity) was calculated and expressed as fold change over the levels of SEP at the first time point after photobleaching. The image data were analyzed with a two-tailed Mann Whitney U test in order to calculate p-values.

## FM4-–64 imaging of primary hippocampal neurons

FM4–64 imaging was performed as described previously (*Gaffield et al., 2006*). Cultures of lentiviral infected primary hippocampal neurons at DIV15 were loaded with 2.5 µM FM4-–64 (Invitrogen) for 2 min in saline solution containing 170 mM NaCl, 3.5 mM KCl, 0.4 mM KH2PO4, 5 mM NaHCO3, 1.2 mM Na2SO4, 1.2 mM MgCl2, 1.3 mM CaCl2, 5 mM glucose, 20 mM N-tris(hydroxymethyl)-methyl-2-aminoethane-sulfonic acid, pH 7.4) supplemented with 55 mM KCl. Neurons were rinsed with saline solution only and then incubated with 2.5 µM FM4-–64 in saline solution. The cells were washed three times with saline solution for a total of 5 min, followed by a wash for 10 min with 1 mM ADVASEP-7 (Sigma-Aldrich) in saline solution. FM4-–64 imaging was performed on a Nikon Ti-E inverted microscope with a Hamamatsu ORCA R2 cooled CCD camera controlled with MetaMorph 7 software with a Plan-Apochromat 40 × 0.95 N.A objective with images taken every 10 s at 25°C. A 1 min baseline was recorded, followed by stimulation with 55 mM KCl in saline solution for 7 min. Cells were excited at 558 nm and the emission measured at 734 nm. Images were analyzed in a blinded manner in ImageJ using the 'time series analyzer v3' plug-in. We analyzed potential functional nerve terminals located along the GFP-positive cells and calculated the average values for each neuron separately and then the average of all neurons per condition. The FM4–64 signal was determined by F = (F1 − B1)/(F0 − B0). The signal was normalized to mean fluorescence intensity measured at baseline condition.

## Measurement of uptake and release of [H$^3$]-glutamate

Primary hippocampal neurons were cultured and transduced with miRZip-153, miRZip-scr, miR-153 or scramble overexpression lentiviruses as described above. For uptake assays, [H$^3$]-glutamate (1 µCi) was added and incubated for 30 min at 37°C. After removal of excess radiolabeled ligands, the cells were washed rapidly three times with ice-cold PBS. The radioactivity remaining in the cells was extracted with NaOH and measured with a liquid scintillation counter. For the release assays, cells were incubated in neurobasal medium with [H$^3$]-glutamate (1 µCi) for 2 hr and washed three times to remove excess radioactivity. The cells were then stimulated as described above (see Stimulation of neuronal cells) for 15 min at 37°C. The media were collected and rapidly centrifuged at 10,000 rpm for 20 s at 4°C. The radioactivity in the supernatants was then measured with a liquid scintillation counter.

Rat hippocampal H19-7/miRZip-153 and H19-7/miRZip-scr cells were seeded into 6-well culture plates coated with poly-L-lysine at a density of $1 \times 10^5$ cells per well and cultured in DMEM supplemented with 10% fetal bovine serum, 100 units/mL penicillin, 100 μg/mL streptomycin, 200 μg/mL G418, and 2 mM glutamine (DMEM-proliferation medium) for 2 days. The cells were grown at 34°C in humidified 5% $CO_2$/95% $O_2$. To initiate differentiation, the cells are incubated at 37°C and placed in N2 medium, which consists of DMEM supplemented with 1% fetal bovine serum, 100 units/mL penicillin, 100 μg/mL streptomycin, 200 μg/mL G418, 2 mM glutamine, and supplemented with 50 ng/mL of IGF-1 (Life Technologies). For the uptake assays, after 48 hr, cells were washed three times with N2 medium containing 31% low $K^+$ buffer (17 mM KCl, 2 mM CaCl2, 1 mM MgCl2, 10 mM HEPES). $[H^3]$-glutamate (1 μCi) was added and incubated for 30 min at 37°C. After removal of excess radiolabeled ligands, the cells were washed rapidly three times with ice-cold PBS. The radioactivity remaining in the cells was extracted with NaOH and measured with a liquid scintillation counter. For the release assays, after 48 hr in N2 medium, cells were incubated with N2 medium containing 31% low $K^+$ buffer (17 mM KCl, 2 mM CaCl2, 1 mM MgCl2, 10 mM HEPES) and $[H^3]$glutamate (1 μCi) for 2 hr and washed three times to remove excess radioactivity. The cells were then stimulated with N2 medium containing 31% low $K^+$ buffer (17 mM KCl, 2 mM CaCl2, 1 mM MgCl2, 10 mM HEPES) or with N2 medium containing 31% high $K^+$ depolarization buffer (170 mM KCl, 2 mM CaCl2, 1 mM MgCl2, 10 mM HEPES) for 30 min at 37°C. The media were collected and rapidly centrifuged at 10,000 rpm for 20 s at 4°C. The radioactivity in the supernatants was then measured with a liquid scintillation counter.

## Data deposition

All miRNA microarray data used in this study have been submitted to the Gene Expression Omnibus (GEO) repository (http://www.ncbi.nlm.nih.gov/geo/) under accession numbers GSE84261 and GSE84262.

## Acknowledgements

We thank Anthony Morielli and members of the Moazed laboratory for helpful discussions, Nahid Iglesias for comments and help with figures, and Phil Williams, Peter Wang, David Strochlic, Erika Williams, Jamal Green, Anthony Hill, Jennifer Waters, Talley Lambert, Hunter Elliott, Nicolas Preitner, Marina Vidaki, the Nikon Imaging Center, and IDAC core at Harvard Medical School for valuable technical assistance. This work was supported by the Damon Runyon Cancer Research Foundation DRG-2042–10 (RSM), an EMBO long-term fellowship (AT), an NARSAD Young Investigator Award (AR), NIH RF1 grant (AG047661) and JPB Foundation grant (L-HT), NIH R01 MH091429 and NS0925578 grants (HU), NIH RO1 DA033123 (MEB), and a grant from Biogen Idec (DM). DM and CAW are Investigators of the Howard Hughes Medical Institute.

## Additional information

### Funding

| Funder | Grant reference number | Author |
|---|---|---|
| Damon Runyon Cancer Research Foundation | DRG-2042–10 | Rebecca S Mathew |
| European Molecular Biology Organization | Long-term fellowship ALTF 197-2009 | Antonis Tatarakis |
| Brain and Behavior Research Foundation | NARSAD Young Investigator Award | Andrii Rudenko |
| Howard Hughes Medical Institute | | Christopher A Walsh Danesh Moazed |
| National Institutes of Health | AG047661 | Li-Huei Tsai |
| JPB Foundation | | Li-Huei Tsai |
| National Institutes of Health | MH091429 | Hisashi Umemori |
| National Institutes of Health | NS0925578 | Hisashi Umemori |

| National Institutes of Health | DA033123 | Mark E Bouton |
| Biogen Idec | | Danesh Moazed |

The funders had no role in study design, data collection and interpretation, or the decision to submit the work for publication.

## Author contributions

RSM, AT, Conceptualization, Formal analysis, Funding acquisition, Investigation, Methodology, Writing—original draft, Writing—review and editing; AR, Conceptualization, Formal analysis, Investigation; EMJ-V, YJY, EAM, TPT, STS, Formal analysis, Investigation; NS, AJM, Investigation; WAF, SEH, Conceptualization, Investigation, Methodology; CAW, Supervision, Funding acquisition, Methodology; L-HT, HU, MEB, Conceptualization, Supervision, Funding acquisition, Methodology; DM, Conceptualization, Formal analysis, Supervision, Funding acquisition, Methodology, Writing—review and editing

## Author ORCIDs

Scott T Schepers, http://orcid.org/0000-0001-8051-7541
Hisashi Umemori, http://orcid.org/0000-0001-7198-2062
Danesh Moazed, http://orcid.org/0000-0003-0321-6221

## Ethics

Animal experimentation: This study was performed in strict accordance with the recommendations in the Guide for the Care and Use of Laboratory Animals of the National Institutes of Health. All of the animals were handled according to approved institutional animal care and use committee (IACUC) protocols of Harvard Medical School. The protocol was approved by the Committee on the Ethics of Animal Experiments of Harvard Medical School. All surgery was performed under sodium pentobarbital anesthesia, and every effort was made to minimize suffering.

## Additional files

### Supplementary files

• Supplementary file 1. (A) List of 353 targets with predicted binding sites for 3 or more of the fear-induced miRNAs. (B) List of targets from the vesicle exocytosis pathway with predicted binding sites for 3 or more fear-induced miRNAs. (C) Predicted Targets of miR-153 from top five Metacore networks.

### Major datasets

The following datasets were generated:

| Author(s) | Year | Dataset title | Dataset URL | Database, license, and accessibility information |
|---|---|---|---|---|
| Rebecca S Mathew, Antonis Tatarakis, Andrii Rudenko, Erin M Johnson-Venkatesh, Yawei J Yang, Elisabeth A Murphy, Travis P Todd, Scott T Schepers, Nertila Siuti, Anthony J Martorell, William A Falls, Sayamwong E Hammack, Christopher A Walsh, Li-Huei Tsai, Hisashi Umemori, Mark E Bouton, Danesh Moazed | 2017 | Adult Female Rat Hippocampal Tissue: Naïve vs. Contextual Fear Conditioned (Trained)[Experiment 1] | https://www.ncbi.nlm.nih.gov/geo/query/acc.cgi?acc=GSE84261 | Publicly available at the NCBI Gene Expression Omnibus (Accession no. GSE84261) |

| Rebecca S Mathew, Antonis Tatarakis, Andrii Rudenko, Erin M Johnson-Venkatesh, Yawei J Yang, Elisabeth A Murphy, Travis P Todd, Scott T Schepers, Nertila Siuti, Anthony J Martorell, William A Falls, Sayamwong E Hammack, Christopher A Walsh, Li-Huei Tsai, Hisashi Umemori, Mark E Bouton, Danesh Moazed | 2017 | Adult Female Rat Hippocampal Tissue: Naïve vs. Contextual Fear Conditioned (Trained)[Experiment 2] | https://www.ncbi.nlm.nih.gov/geo/query/acc.cgi?acc=GSE84262 | Publicly available at the NCBI Gene Expression Omnibus (Accession no. GSE84262) |

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
