## [Decision Letter]

[Editors’ note: a previous version of this study was rejected after peer review, but the authors submitted for reconsideration. The first decision letter after peer review is shown below.]

Thank you for submitting your work entitled "Inhibition of Memory by a microRNA Negative Feedback Loop that Downregulates SNARE-Mediated Vesicle Transport" for consideration by *eLife*. Your article has been reviewed by two peer reviewers, and the evaluation has been overseen by Gary Westbrook as the Senior Editor and Reviewing Editor. The reviewers have opted to remain anonymous. Our decision has been reached after consultation between the reviewers. Based on these discussions and the individual reviews below, we regret to inform you that your work will not be considered further for publication in *eLife*.

This manuscript reflects a creative and collaborative approach between top scientists in behavior, RNA biology, plasticity and brain development. The reviewers were enthusiastic about the topic and the general hypothesis, but had serious concerns about how the many pieces fit together, given disconnect in preparations and time points examined. The overall sense was that this work has the potential to be influential but that it would require substantial more data to fix the inconsistencies and gaps. For example, the authors do not have strong functional data demonstrating altered "Snare-mediated vesicle transport" and that altering miR-153 has a physiological impact on vesicles. Because *eLife* policy is not to consider manuscripts that require more than modest revisions, we are returning the manuscript to you at this point. If after further work, the authors want to consider another submission to *eLife* we encourage you to contact the editors.

*Reviewer #1:*

"Inhibition of Memory by a microRNA Negative Feedback Loop that Downregulates SNARE-Mediated Vesicle Transport" takes a multi-disciplinary approach to the discovery of microRNA's regulated by learning behaviors and the physiological role of one of these microRNAs, miR-153, is explored. The overall story is interesting and of broad significance. I find the gene regulation data convincing. However, the data supporting behavioral impact and a physiologically relevant regulation vesicle fusion/transport by miR-153 needs to be strengthened to truly justify the major conclusions of the paper. In addition, there are some pieces of data that appear internally inconsistent. I believe these detract from the manuscript.

1) For the network analysis in Figure 1. The list of predicted targets for miR-153 was narrowed by requiring that the target be expressed in the brain. Thus, the sample would be biased at this point towards brain specific functions. Thus, to determine if the predicted targets are indicative of a specific function there needs to be a comparison to a random set of brain-expressed genes.

2) In Figure 1 and Figure 1—figure supplement 1, the in situ data appears to show an increase in staining in the whole hippocampus after training. In particular, the stratum oriens of the CA1 appears to have as intense an increase in miR-153 staining as does the DG. This is inconsistent with the conclusion that regulation is dentate specific.

3) In Figure 3—figure supplement 1. The images of the dentate gyrus infected with the miR-153 knockdown lentivirus appear to show expression in about 10-20% of the granule neurons (A). This draws concerns about any behavioral changes that could be elicited by knockout of miR-153 in such a small percentage of the total granule cell population. Quantification of miR153 levels in the FACS sorted cells demonstrates good control of miR-153 by the virus in infected cells. However, an indication of the percentage of the cells infected or the overall decrease in miR-153 in the dentate of animals used for behavioral testing is needed. A plot of the correlation of freezing time in the contextual fear conditioning versus percentage of infected cells or overall expression of miR-153 could potentially strengthen the argument that the behavioral change is truly the result of miR-153 manipulation.

4) In the subsection “miR-153 Negatively Regulates Spine Size” it is stated that there was an increase in spine head width after knockdown of 153. This is not supported by the statistical analysis.

5) The data demonstrating that the cellular function of miR153 regulates vesicle fusion is lacking. This should be more thoroughly addressed in neurons. The lack of any knockdown phenotype is a concern. Given the induction of miR-153 by potassium I would suggest morphological analysis of these cultures in highK after miR-153 knockdown.

6) The only data indicating altered vesicle fusion/transport is panels D and E in Figure 7 which come from the hippocampal cell line. While the gene expression data support the claims in the title, to truly substantiate these claims there needs to be stronger support for a functional impact on SNARE-mediated vesicle transport/fusion. Given that you have a system to manipulate miR-153 in hippocampal cultures or in vivo I suggest that there needs to be experiments demonstrating alteration of presynaptic function in one of these systems.

*Reviewer #2:*

This is an impressive study from the Moazad lab, with input from the labs of Li-Huai Tsai, Chris Walsh and Mark Bouton, that describes a role for miRNAs, and in particular for miR-153, during fear-conditioning learning and synaptic plasticity. The manuscript describes a large number of studies, including behavioral studies in rat, long-term potentiation studies in mouse hippocampal slices, and biochemical/molecular studies in dissociated mouse neurons. Based on the results of the experiments, the authors argue that miRNA-mediated post-transcriptional regulation serves as a brake on plasticity, preventing activity-induced neuronal hyperexcitability. Together, the study aims to provide a compelling example of the importance of miRNA-mediated post-transcriptional regulation during adult plasticity, learning and memory. Although each set of experiments is well-done, specific hypotheses are tested in distinct preparations at distinct time points, and the results are subsequently woven together to create a story. This reduces my enthusiasm for the study. I would have liked to see, for example, that the targets of miR-153 are in fact down-regulated in dentate gyrus granule cells following fear conditioning. I also had a number of questions about the cell biology of the results, and specifically about what the effects on vesicular transport in presynaptic as opposed to postsynaptic compartments.

---

## [Author Response]

[Editors’ note: the author responses to the first round of peer review follow.]

*This manuscript reflects a creative and collaborative approach between top scientists in behavior, RNA biology, plasticity and brain development. The reviewers were enthusiastic about the topic and the general hypothesis, but had serious concerns about how the many pieces fit together, given disconnect in preparations and time points examined. The overall sense was that this work has the potential to be influential but that it would require substantial more data to fix the inconsistencies and gaps. For example, the authors do not have strong functional data demonstrating altered "Snare-mediated vesicle transport" and that altering miR-153 has a physiological impact on vesicles. Because eLife policy is not to consider manuscripts that require more than modest revisions, we are returning the manuscript to you at this point. If after further work, the authors want to consider another submission to eLife we encourage you to contact the editors.*

We have performed a considerable number of additional experiments that address the concerns summarized above and would like to submit a revised manuscript for consideration.

*Reviewer #1:*

*"Inhibition of Memory by a microRNA Negative Feedback Loop that Downregulates SNARE-Mediated Vesicle Transport" takes a multi-disciplinary approach to the discovery of microRNA's regulated by learning behaviors and the physiological role of one of these microRNAs, miR-153, is explored. The overall story is interesting and of broad significance. I find the gene regulation data convincing. However, the data supporting behavioral impact and a physiologically relevant regulation vesicle fusion/transport by miR-153 needs to be strengthened to truly justify the major conclusions of the paper. In addition, there are some pieces of data that appear internally inconsistent. I believe these detract from the manuscript.*

*1) For the network analysis in Figure 1. The list of predicted targets for miR-153 was narrowed by requiring that the target be expressed in the brain. Thus, the sample would be biased at this point towards brain specific functions. Thus, to determine if the predicted targets are indicative of a specific function there needs to be a comparison to a random set of brain-expressed genes.*

We thank the reviewer for this insightful comment. As suggested by the reviewer, to determine whether the predicted targets from the 21 fear-induced miRNAs are indicative of a specific function, we performed a comparison to random sets of brain-expressed genes. We obtained a list of ~20,000 brain-expressed genes (Lein et al., 2007) and used this database to generate random sets of genes for network analysis. Random sets of genes were created that were similar in size to the target gene list (3,986) and network analysis was performed with each random gene list.Figure 1 has been revised to include an additional panel with statistical significance (p-value) for each canonical signaling pathway as compared to the number of occurrences from random sets of brain-expressed genes (p-value column). As indicated in the revised Figure 1, all 6 of the networks gave p-values of <0.0001; the Synaptic Vesicle Exocytosis, Transmission of nerve impulse and Long-term potentiation networks gave p-values <10^-24^.

*2) In Figure 1 and Figure 1—figure supplement 1, the in situ data appears to show an increase in staining in the whole hippocampus after training. In particular, the stratum oriens of the CA1 appears to have as intense an increase in miR-153 staining as does the DG. This is inconsistent with the conclusion that regulation is dentate specific.*

As the reviewer correctly points out, we observed an increase in staining throughout the hippocampus after training in our in situ hybridization experiments. These data contradict our RT-qPCR results, which clearly and reproducibly demonstrated that miR-153 expression was increased specifically in the DG after fear conditioning. We believe that this inconsistency is due to non-specific background staining in the in situ hybridization experiments, as high background was also observed with a scrambled negative control probe. Because we are confident in the RT-qPCR results, we have removed Figure 1, Figure 1—figure supplement 1, and the primary culture in situ hybridization data from Figure 5—figure supplement 2, and will present only the RT-qPCR data. We note that the RT-qPCR data fully support our conclusion that expression of miR-153 is primarily induced in the hippocampal DG region (Figure 3).

*3) In Figure 3—figure supplement 1. The images of the dentate gyrus infected with the miR-153 knockdown lentivirus appear to show expression in about 10-20% of the granule neurons (A). This draws concerns about any behavioral changes that could be elicited by knockout of miR-153 in such a small percentage of the total granule cell population. Quantification of miR153 levels in the FACS sorted cells demonstrates good control of miR-153 by the virus in infected cells. However, an indication of the percentage of the cells infected or the overall decrease in miR-153 in the dentate of animals used for behavioral testing is needed. A plot of the correlation of freezing time in the contextual fear conditioning versus percentage of infected cells or overall expression of miR-153 could potentially strengthen the argument that the behavioral change is truly the result of miR-153 manipulation.*

FACS sorting of hippocampal cells revealed 10% of the total cell population in the hippocampus was GFP+, from a total of 900,000 cells that were viable at the time of FACS sorting. Although the total number of cells is reduced relative to the quantity expected from the entire hippocampus, this value provides a minimum percentage of cells infected with the miR-153 or scramble control lentivirus. We have revised our Methods section to include the percentage of the cells infected with the miR-153 knockdown lentivirus in the DG. Long-term gene knockdown with the use of lentiviral vectors delivering shRNA has been successfully performed previously, for example, see Abbas- Terki et al., 2002; Chew et al., 2015; Seifert et al., 2015; Gao et al., 2010; Kanninen et al., 2009; Goshen et al., 2011. Our lentiviral- mediated delivery of shRNA against miR-153 into the mouse DG was intended to reduce miR-153 levels within DG granule cells. In our study, GFP^+^ hippocampal cells showed a significant reduction in miR-153 expression relative to GFP^+^ cells isolated from the scramble control (Figure 5—figure supplement 1). Infection of 10% of the total cell population in the hippocampus was sufficient to elicit behavioral differences between both groups of animals. Our results suggest that increased neurotransmitter release and enhanced vesicle transport, upon knockdown of miR-153, are responsible for the behavioral adaptations observed. In future studies we will assess the impact of miR-153 expression on freezing time in the contextual fear conditioning paradigm but feel that these additional experiments are better suited for a separate manuscript.

*4) In the subsection “miR-153 Negatively Regulates Spine Size” it is stated that there was an increase in spine head width after knockdown of 153. This is not supported by the statistical analysis.*

We agree with the reviewer that results were not presented clearly. To test the effects of miR-153 perturbation on dendritic spine morphology and density and present the data, we used two different methods that have been used in the literature before. We calculated the mean values of each feature of the spine for every individual neuron separately and then plotted the average of all neurons per condition (Figure 6 and Figure 6—figure supplement 1). Additionally, we presented the data as cumulative plots including all the measurements of the individual spines, using similar numbers of spines per individual neuron for each condition (Figure 6—figure supplement 1). Spine head width based on the calculation of the mean values had an increase of 0.04μm (stated in the main text), which was not significant compared to control cells. However, in the cumulative plots using a Kolmogorov-Smirnov test to calculate statistical significance, we observed increase in the spine head width in the miR-153KD neurons compared to the scramble KD, with a p value p<0.001 and D=0.09, as we stated in the legend of Figure 6—figure supplement 1. Since spine head width has a range of different values, presenting the data in the form of cumulative plots allows for detection of moderate changes. For example, changes that occur in a group of spines of a particular size, which would be missed otherwise. In the revised version of the manuscript, we modified the text to further clarify this point.

*5) The data demonstrating that the cellular function of miR153 regulates vesicle fusion is lacking. This should be more thoroughly addressed in neurons. The lack of any knockdown phenotype is a concern. Given the induction of miR-153 by potassium I would suggest morphological analysis of these cultures in highK after miR-153 knockdown.*

We thank the reviewer for this suggestion and have employed two strategies to test the role of miR-153 in vesicle transport in primary neurons. First, we used an FM dye-based assay widely used to image synaptic vesicle exocytosis (Gaffield et al., 2006). Specifically, we performed lentiviral-mediated inhibition of miR-153 in primary hippocampal neurons and used the FM4-64 dye to label and visualize their synaptic vesicles. We found that KCl-induced depolarization of the hippocampal neurons promoted exocytosis of the labeled vesicles at a higher rate in the miR-153KD neurons compared to control neurons, as indicatedby the faster depletion of FM4-64 dye.

Additionally, overexpression of miR-153 in hippocampal neurons resulted in the opposite effect, decreasing the rate of vesicle exocytosis. These new data support our conclusion for the role of miR-153 in regulation of vesicle exocytosis and are depicted in new Figure 9 (panels A and B) of the revised manuscript. Second, we assessed the effect of miR-153 knockdown and overexpression on neurotransmitter release in primary hippocampal cultures following neuronal activation using the biochemical assay that we had used in a cell line in the previous version of the manuscript. We observe increased neurotransmitter release after lentiviral-mediated miR-153 knockdown, whereas miR-153 overexpression result in lower levels of neurotransmitter release compared to control neurons (new Figure 9). Therefore, we have confirmed the effects we observed previously in the hippocampal cell line, using this same assay, in primary cultured neurons. As expected, the results further show that miR-153 overexpression has the opposite effect.

Overall, this new data strengthens our claim that miR-153 regulates synaptic vesicle exocytosis. Using two different approaches in primary hippocampal neurons, we demonstrated the key role of miR-153 in vesicle exocytosis.These new data are presented in Figure 9 and replace the data in the previous version of the manuscript from cell lines (now moved to Figure 9—figure supplement 1).

*6) The only data indicating altered vesicle fusion/transport is panels D and E in Figure 7 which come from the hippocampal cell line. While the gene expression data support the claims in the title, to truly substantiate these claims there needs to be stronger support for a functional impact on SNARE-mediated vesicle transport/fusion. Given that you have a system to manipulate miR-153 in hippocampal cultures or* in vivo *I suggest that there needs to be experiments demonstrating alteration of presynaptic function in one of these systems.*

We thank the reviewer for this comment. As suggested by the reviewer, we examined the effect of knockdown and overexpression of miR-153 on neurotransmitter release in primary hippocampal cultures after stimulation. We observe altered presynaptic function after both knockdown and overexpression of miR-153, compared to knockdown and overexpression of a scrambled control. These findings are presented in Figure 9. We do not observe any effect on neurotransmitter uptake after spontaneous activity (low KCl conditions), supporting our conclusion that miR-153 acts to suppress neurotransmitter release following a period of stimulation (high KCl conditions). The neurotransmitter uptake data as well as the results from the hippocampal cell line are presented in Figure 9—figure supplement 1.

*Reviewer #2:*

*This is an impressive study from the Moazad lab, with input from the labs of Li-Huai Tsai, Chris Walsh and Mark Bouton, that describes a role for miRNAs, and in particular for miR-153, during fear-conditioning learning and synaptic plasticity. The manuscript describes a large number of studies, including behavioral studies in rat, long-term potentiation studies in mouse hippocampal slices, and biochemical/molecular studies in dissociated mouse neurons. Based on the results of the experiments, the authors argue that miRNA-mediated post-transcriptional regulation serves as a brake on plasticity, preventing activity-induced neuronal hyperexcitability. Together, the study aims to provide a compelling example of the importance of miRNA-mediated post-transcriptional regulation during adult plasticity, learning and memory. Although each set of experiments is well-done, specific hypotheses are tested in distinct preparations at distinct time points, and the results are subsequently woven together to create a story. This reduces my enthusiasm for the study. I would have liked to see, for example, that the targets of miR-153 are in fact down-regulated in dentate gyrus granule cells following fear conditioning. I also had a number of questions about the cell biology of the results, and specifically about what the effects on vesicular transport in presynaptic as opposed to postsynaptic compartments.*

We thank the reviewer for insightful comments. To address the reviewer’s concern about down-regulation of miR-153 targets in the hippocampus, we performed RT-qPCR experiments for the six SNARE-mediated vesicle exocytosis targets of miR-153 (characterized in Figure 8) using the same hippocampal RNA isolated from naïve and trained rats, which we had used for quantifying changes in miRNA levels. We found that five of the six target genes (*Pclo, Snca, Snap25, Trak2*, and *Vamp2*) were downregulated 24 hours following fear conditioning. These results are presented in the revised manuscript in Figure 8 and strengthen our conclusion that the vesicular transport pathway is a major target for downregulation during fear conditioning. In future studies we will assess the impact of miR-153 manipulation on downstream targets in dentate gyrus granule cells after fear conditioning but feel that these experiments are better suited for a separate manuscript.